# A conserved microtubule-binding region in *Xanthomonas* XopL is indispensable for induced plant cell death reactions

Simon Ortmann[1,2☺], Jolina Marx[1☺¤], Christina Lampe[1], Vinzenz Handrick[2], Tim-Martin Ehnert[2], Sarah Zinecker[3], Matthias Reimers[3], Ulla Bonas[1], Jessica Lee Erickson[1,2¤]*

**1** Department of Genetics, Institute for Biology, Martin Luther University Halle-Wittenberg, Halle, Germany,
**2** Department of Biochemistry of Plant Interactions, Leibniz Institute for Plant Biochemistry, Halle, Germany,
**3** Department of Plant Physiology, Institute for Biology, Martin Luther University Halle-Wittenberg, Halle, Germany

☺ These authors contributed equally to this work.
¤ Current address: Department of Biochemistry of Plant Interactions, Leibniz Institute for Plant Biochemistry, Halle, Germany
* jessica.erickson@ipb-halle.de

**Data Availability Statement:** All relevant data are within the manuscript and its Supporting Information files.

## Abstract

Pathogenic *Xanthomonas* bacteria cause disease on more than 400 plant species. These Gram-negative bacteria utilize the type III secretion system to inject type III effector proteins (T3Es) directly into the plant cell cytosol where they can manipulate plant pathways to promote virulence. The host range of a given *Xanthomonas* species is limited, and T3E repertoires are specialized during interactions with specific plant species. Some effectors, however, are retained across most strains, such as Xanthomonas Outer Protein L (XopL). As an 'ancestral' effector, XopL contributes to the virulence of multiple xanthomonads, infecting diverse plant species. XopL homologs harbor a combination of a leucine-rich-repeat (LRR) domain and an XL-box which has E3 ligase activity. Despite similar domain structure there is evidence to suggest that XopL function has diverged, exemplified by the finding that XopLs expressed in plants often display bacterial species-dependent differences in their sub-cellular localization and plant cell death reactions. We found that XopL from *X. euvesicatoria* (XopL$_{Xe}$) directly associates with plant microtubules (MTs) and causes strong cell death in agroinfection assays in *N. benthamiana*. Localization of XopL$_{Xe}$ homologs from three additional *Xanthomonas* species, of diverse infection strategy and plant host, revealed that the distantly related *X. campestris* pv. *campestris* harbors a XopL (XopL$_{Xcc}$) that fails to localize to MTs and to cause plant cell death. Comparative sequence analyses of MT-binding XopLs and XopL$_{Xcc}$ identified a proline-rich-region (PRR)/α-helical region important for MT localization. Functional analyses of XopL$_{Xe}$ truncations and amino acid exchanges within the PRR suggest that MT-localized XopL activity is required for plant cell death reactions. This study exemplifies how the study of a T3E within the context of a genus rather than a single species can shed light on how effector localization is linked to biochemical activity.

**Funding:** The salaries of JLE and VH were paid through awards from the Deutsche Forschungsgemeinschaft (DFG) (Walter Benjamin Fellowships awarded to JLE, ER 1024/1-1 and VH, HA 8855/1-1). Work was also funded by the DFG grant CRC 648 "Molecular mechanisms of information processing in plants". A portion of JLE's salary and the full salary of CL was paid by the Gottfried Wilhelm Leibniz-Prize awarded to UB, BO 790/9-1. The salary of UB was paid by Martin/ 12-1. The salary of UB was paid by Martin Luther University. Core funding of the Leibniz Institute for Plant Biochemistry (IPB) paid the salary of TME and SO. JM, SZ and MR received no specific funding for this work. The funders had no role in study design, data collection and analysis, decision to publish, or manuscript preparation.

**Competing interests:** The authors have declared that no competing interests exist.

## Author summary

Xanthomonas Outer Proteins (Xops) are type III effector proteins originating from bacterial plant pathogens of the *Xanthomonas* genus. *Xanthomonas* uses a needle-like structure to inject a cocktail of Xops directly into plant cells where they manipulate cellular processes to promote virulence. Previous studies of individual Xops have provided valuable insights into virulence strategies used by *Xanthomonas*, knowledge that can be exploited to fight plant disease. However, despite rapid progress in the field, there is much about effector activity we still do not understand.

Our study focuses on the effector XopL, a protein with E3 ligase activity that is important for *Xanthomonas* virulence. In this study we expressed XopLs in leaves of the model plant *N. benthamiana* and found that XopLs from different *Xanthomonas* species differ in their subcellular localization. XopLs from closely related species associate with the microtubule cytoskeleton and disassemble it, whereas a XopL from a distantly related species did not. This prompted a comparative analysis of these proteins, which showed how microtubule binding is achieved and how it affects the plant response to XopL.

## Introduction

*Xanthomonas* is a genus of Gram-negative phytopathogens that threaten more than 400 plant species worldwide, causing disease on crop plants such as bacterial leaf spot, citrus canker, bacterial blight, black rot and Xanthomonas wilt [1, 2]. The bacteria gain access to plant tissue through wounds or natural openings provided by hydathodes or stomata and, depending on the pathogen species, colonize the plant vascular tissue or the intercellular spaces between mesophyll cells [2]. While initially acquiring nutrients and proliferating in live tissue (biotrophic phase), *Xanthomonas* infections eventually cause cell death (necrotrophic phase) in susceptible plant genotypes, giving rise to visible necrotic spots or lesions [3].

Like most pathogenic diderm bacteria that infect eukaryotes, the majority of *Xanthomonas* strains possess a type III secretion system (T3SS) to translocate type III effector proteins (T3Es) directly into the plant cytosol [4, 5]. Once inside the cell, T3Es manipulate plant cells in favor of the bacteria, e.g., by suppressing basal immune responses and promoting sugar export [6]. T3E interference in host cell processes is achieved through an incredibly diverse array of biochemical activities, including, but not limited to, transcription activation, uridylation, ubiquitination and phosphorylation of host proteins [1, 6]. Over the last 25 years the biochemical characterization of individual T3Es and their interactions with plant targets has shed light on virulence strategies employed by bacterial pathogens, thus exposing plant pathways that are vulnerable to infections.

T3Es translocated by different *Xanthomonas* species (most of which are termed *Xanthomonas* outer proteins; Xops) vary in number and biochemical activity, as effector arsenals have evolved through interactions with specific host species [2, 7]. For example, *Xanthomonas euvesicatoria* (*Xe*) strain 85–10 (formerly known as *Xanthomonas campestris* pv. *vesicatoria* [8]), the causal agent of bacterial spot disease on tomato and pepper, secretes at least 36 effector proteins [9], whereas *Xanthomonas campestris* pv. *campestris* (*Xcc*) strain 8004, a Brassicaceae pathogen, expresses around 24 predicted T3Es [10]. Despite between-species variation, there is a subset of ancestral T3Es that are present in the majority of *Xanthomonas* strains, including XopL [7, 11], which was first isolated from *Xe* 85–10 and identified as an E3 ligase by Singer et al. [12]. As its retention among xanthomonads suggests, XopL significantly contributes to

the virulence of *Xe* 85–10 on tomato [13], of *X. axonopodis* pv. *punicae* (*Xap*) on pomegranate [14] and of *Xcc* 8004 on Chinese radish [15] and Chinese cabbage [16]. XopL from *Xe* 85–10 (XopL$_{Xe}$) interacts with and hijacks the plant ubiquitination machinery [12], post-translationally modifies host proteins and signals their degradation [13]. Although XopL mimics the activity of plant E3 ligases, the 3D structure revealed that the C-terminal ligase domain (XL-box) was structurally unique to all previously characterized E3 ligases [12]. XopL also harbors an N-terminal disordered region (containing the type III secretion signal) followed by three alpha helices and a leucine-rich-repeat (LRR) domain that is hypothesized to be important for the recognition of substrate proteins [12]. XopL homologs consistently carry a LRR and an XL-box with conserved folds, but the sequence and length of their N-termini are highly variable between bacterial species [12]. Previous studies utilizing the expression of fluorescently tagged XopLs suggest surprising strain-dependent variability in the sub-cellular localization of XopLs. For example, XopL$_{Xoo}$ (from *X. oryzae* pv. *oryzae*) was primarily cytosolic [17], *Xap* XopL (XopL$_{Xap}$) was at the plasma membrane [14], and XopL$_{Xe}$ was found in the nucleus, cytosol [18] and, most recently, in autophagosomes [13]. Furthermore, an E3 ligase-inactive variant of XopL$_{Xe}$ with triple amino acid exchanges in the XL-box (H584A, L585A, G586E, herein referred to as XopL$_m$) clearly labeled plant microtubules (MTs) [18]. Species-dependent differences can also be observed in the examination of macroscopic phenotypes resulting from *Agrobacterium tumefaciens*-mediated expression of XopL in the leaves of *Nicotiana benthamiana*. Plant cell death has been previously used as a read-out for enzymatic activity of XopL proteins from both *Xe* 85–10 and the rice pathogen *Xoo* PX099A (74.24% amino acid identity) [12, 17]. In the case of XopL$_{Xe}$, cell death was shown to rely on the presence of both the LRR and active XL-box, suggesting that the modification of a specific plant target is responsible for this phenotype [12]. By contrast, similar experiments with XopL from *Xcc* 8004 (XopL$_{Xcc}$) did not result in a plant cell death reaction [16].

The apparent variability in XopL subcellular localization in plant cells and differences in cell death phenotypes induced by XopLs of different xanthomonads hint that, despite being retained as an ancestral T3E with conserved structural features (LRR and XL-box), XopL function has diverged. Previous studies of XopL have focused on the identification of protein interactors to elucidate its actions within the plant cell [13, 17]. However, we employed an alternative approach and analyzed natural variation in XopL proteins between *Xanthomonas* strains to gain insight into how differences in protein localization are achieved and whether they are linked to effector functionality. XopL$_{Xcc}$ and XopL$_{Xe}$ were chosen as a case study because they originate from distantly related *Xanthomonas* species which differ in their infection strategy (vascular vs. non-vascular) and their host plants, both measurably contribute to virulence, and they differ in their ability to induce cell death *in planta*.

## Results

*A. tumefaciens*-mediated protein expression (coined 'agroinfection') in the *N. benthamiana* model system was chosen as a platform to study XopL, as it enables rapid and reliable transgene expression (1–3 days), in contrast to natural *Xanthomonas* hosts like tomato, pepper, or Brassicaceae. *N. benthamiana* is one of the most common plant models used for T3E research, which also allows for the direct comparison of our work with that of other groups. While T3E protein levels are typically more abundant during transient assays than during an infection with *Xanthomonas*, the use of agroinfection eliminates many extraneous variables that complicate the study of individual effectors, e.g., interactions with other T3Es and immune responses triggered by *Xanthomonas*. In addition, agroinfection to express translational fusions of T3Es with fluorescence proteins facilitates the microscopic examination of effector localization

within plant cells. This strategy is used as an alternative to expressing tagged effectors in the native *Xanthomonas*, since large fluorescence tags are not translocated through the T3SS [19].

## Cell death is induced by the expression of XopL$_{Xe}$ but not XopL$_{Xcc}$

As mentioned, the activity of XopL from *Xe* 85–10 (XopL$_{Xe}$) is easily monitored in the context of transient expression assays due to a robust cell death reaction that is visible 5–7 days post inoculation (dpi); a reaction that requires a functional E3 ligase domain (XL-box) plus the LRR, the putative protein interaction domain [12]. Using the same model system it was independently reported that cell death was not induced by XopL from *Xcc* 8004 (XopL$_{Xcc}$) [16], a first indication of possible divergence in XopL activity between different xanthomonads. To confirm these results in our experimental conditions, translational XopL-GFP fusions were generated and expressed side-by-side in *N. benthamiana* leaves. Cell death was recorded at 5 dpi (Fig 1A); our results confirmed the results of Singer et al. [12] and Yan et al. [16]. We observed a strong cell death reaction in tissue overexpressing XopL$_{Xe}$ but not XopL$_{Xcc}$, despite similar protein levels (Fig 1B). This supports the hypothesis that the two XopL homologs have different activities *in planta*.

## Enzymatic activity of four XopLs is conserved

As plant cell death induced by XopL$_{Xe}$ is known to depend on its E3 ligase activity [12], we wondered whether XopL$_{Xe}$ and XopL$_{Xcc}$ differ in enzymatic activity. For this, ubiquitination assays were performed with XopL$_{Xcc}$ and XopL$_{Xe}$. As a general test for E3 ligase conservation, XopLs from two additional species were also included: XopL from *X. citri* pv. *citri* (XopL$_{Xac}$), which causes citrus canker disease and is closely related to *Xe* (non-vascular pathogens), and the more distantly related *Xoo* (XopL$_{Xoo}$), the causal agent of leaf blight on rice (vascular/systemic pathogen). *xopL* coding sequences from *Xoo* strain PX099A and *Xac* strain 306 were synthesized and codon-optimized for *N. benthamiana*. For the sake of comparability (see Materials and Methods), XopL$_{Xe}$ and XopL$_{Xcc}$ codon optimized versions were also synthesized for this experiment. E3 ligase activity was tested in the UbiGate system, a modular system for the expression of the plant ubiquitination machinery in *E. coli* [20]. Using UbiGate one can screen for E3 ligase activity of a protein of interest without any protein purification steps. The ability of XopL$_{Xe}$ to efficiently generate polyubiquitin chains *in vitro* was previously demonstrated by Singer et al [12]. In this publication the LRR was found to be dispensable for E3 ligase activity *in vitro* and only the XL-box is required. As a proof of concept, we tested several XopL variants for their ability to generate ubiquitin chains in UbiGate: wild-type XopL, XopL$_m$, a variant including the N-terminus, α-helical region and the LRR (NTαLRR, 1–450 aa) and a truncation with the XL-box alone (451–660 aa). UbiGate assays showed that when co-expressed with the plant ubiquitination machinery in *E. coli* (ubiquitin, E1 and E2 enzymes; plasmid map in S1A Fig) XopL$_{Xe}$-dependent ubiquitin chain formation is observed, as evidenced by laddering of ubiquitin in samples with XopL$_{Xe}$ and this laddering is not observed in the E2- or Ub- controls (S1B Fig). Just as in experiments by Singer et al., [12], expression of the XL-box also results in extensive ubiquitin laddering, while this does not occur in case of XopL$_m$ or NTαLRR (S1B Fig). Interestingly, XopL$_m$ appears to have some minimal residual enzymatic activity, indicated by a small smear in the ubiquitin blot between 100–180 kDa. The residual E3-ligase activity detected seems to occur mostly as autoubiquitination, as the smear is also visible in the α-myc blot (S1C Fig). XopL$_m$ expression does not lead to the formation of free ubiquitin chains in lower molecular weight range (15–40 kDa) or high molecular weight ubiquitin chains which occur in the presence of the fully active wild-type and XL XopL derivatives (S1B Fig). Comparison of the results derived from the UbiGate system and from *in vitro* assays performed with purified XopL protein [12] confirmed that the UbiGate system is

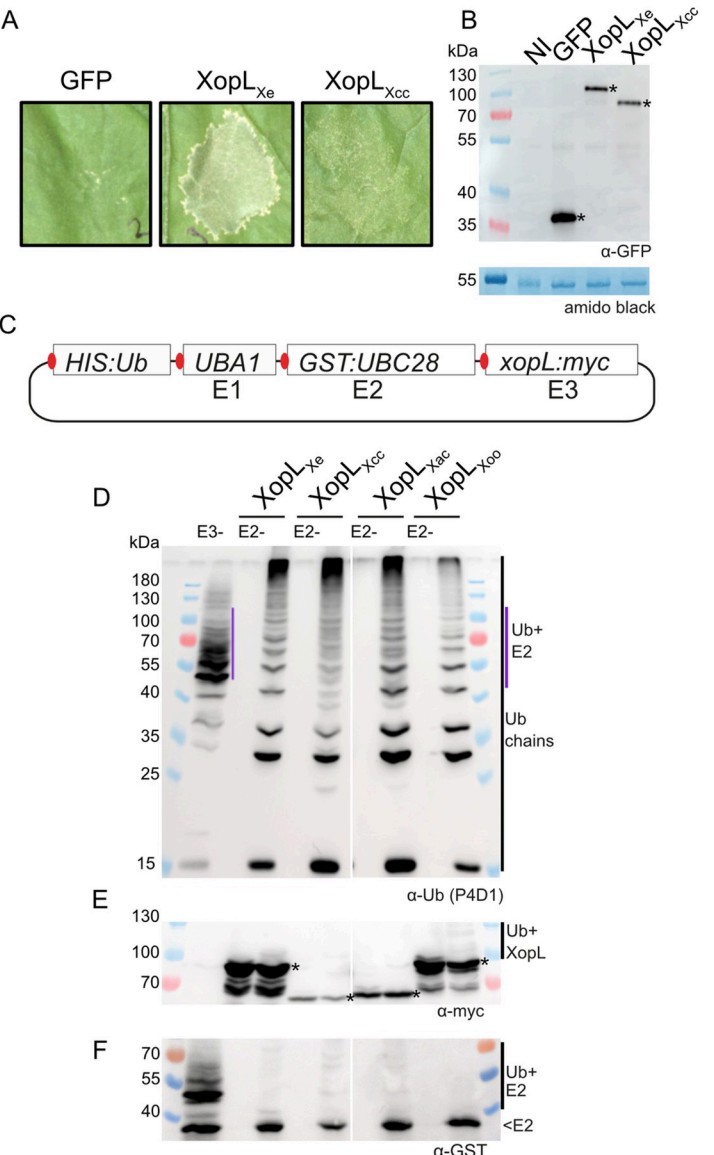

**Fig 1. E3 ligase activity is conserved among XopL homologs from different xanthomonads.** (A) Agroinfection of *N. benthamiana* leaves to express GFP-myc and XopL proteins from *Xanthomonas* strains *Xe* 85–10 (XopL$_{Xe}$) and *Xcc* 8004 (XopL$_{Xcc}$) fused to a C-terminal GFP (OD of 0.8). Plant reactions were monitored 5 dpi. (B) Western blot analysis of protein extracts isolated 2 dpi from samples depicted in (A). One 'not inoculated' (NI) sample was included as a negative control. Signals were detected using a GFP-specific antibody (*). The left side of the blot shows protein mass in kDa. Amido black staining is shown as a loading control. (C) Schematic of the UbiGate plasmid used to test for E3 ligase activity of XopL proteins. Components of the *A. thaliana* ubiquitination machinery (affinity-tagged for detection by western blot), *UBIQUITIN 10* (*HIS:Ub*), *UBIQUITIN ACTIVATING ENZYME 1* (*UBA1*) and *UBIQUITIN CONJUGATING ENZYME 28* (*GST:UBC28*) were expressed from a single, IPTG-inducible plasmid together with *xopL* coding sequences. Each translational unit was equipped with an independent ribosome binding site (red). (D-F) Western blot analysis of protein extracts isolated from *E. coli* expressing different XopL proteins. Controls were samples lacking the E3 ligase (E3-) or the E2 enzyme (E2-). The XopL protein tested is indicated above the designated lanes. (D) Ubiquitin was detected using the P4D1 antibody. Polyubiquitin chains are indicated by a black line. E2-ubiquitin (Ub+E2) conjugates are visible in E3- control sample (purple line). (E) C-terminally tagged XopL proteins were detected with a myc-specific antibody (expected size indicated with '*'). In some cases, autoubiquitination is detectable (Ub+XopL). (F) N-terminally-tagged E2 enzyme was detected using a GST-specific antibody. Samples were run on different gels to clearly visualize ubiquitin and other proteins. Protein mass is expressed in kDa.

suitable for the analysis of XopL E3 ligase activity. Having validated our approach, UbiGate was then used to test for E3 ligase activity of XopLs from the four xanthomonads and revealed that they were all enzymatically active (ubiquitin ladder in all XopL-containing samples; Fig 1D). While this was already known for the closely related $XopL_{Xoo}$ [17], this is new for $XopL_{Xac}$ and $XopL_{Xcc}$. The UbiGate results also suggest that E3 ligase function is conserved, even among distantly related XopLs, and that differences in phenotypes of plant tissue expressing $XopL_{Xe}$ and $XopL_{Xcc}$ are unlikely due to differences in enzymatic activity.

## $XopL_{Xe}$ tag position impacts cell death

Next, we examined the *in planta* subcellular localization of XopL homologs. As mentioned above, localization studies have independently shown that XopL proteins from different *Xanthomonas* species can differ in their localization [13, 14, 17, 18], offering a possible explanation for differences in XopL-mediated plant phenotypes.

$XopL_{Xe}$ was chosen as a starting point for in-depth localization experiments because the functionality of different florescence protein fusions could be easily evaluated using cell death as a readout which is readily observed when XopL is tagged with a C-terminal GFP (Fig 1A). To ensure that localization did not differ depending on tag position, N-terminal fusions were also tested. C- and N-terminal GFP fusions to $XopL_{Xe}$ were tested side-by-side with the untagged XopL protein, as well as myc-GFP and the E3 ligase mutant ($XopL_m$) as negative controls (Fig 2A). These experiments resulted in strong cell death reactions at 5 dpi in tissue expressing untagged XopL or C-terminally-tagged $XopL_{Xe}$. However, cell death was severely reduced in case of an N-terminal fusion. As expected, expression of $XopL_m$ and GFP controls showed no visible cell death. Although reduced cell death by the GFP-XopL fusion could be due to reduced protein level (Fig 2B), the difference in phenotype to untagged XopL might also indicate reduced/altered activity of GFP-$XopL_{Xe}$.

## $XopL_{Xe}$ labels MTs

Confocal microscopy of $XopL_{Xe}$-GFP fusions expressed in *N. benthamiana* leaves showed that, as previously reported [18], the C-terminal fluorescence protein fusion localizes to both mobile dots (blue arrows), as well as to the nucleus and cytosol (Fig 2C), a pattern that the N-terminal fusion also shows (Fig 2E). The expression of both N- and C-terminal XopL fusions also induced the typical plastid clustering phenotype that we previously characterized in detail [18]. The E3 ligase mutant, $XopL_m$, also localized to the nucleus and cytosol but additionally labeled filaments (Fig 2F and 2G). Both C- and N-terminal tagged $XopL_m$ localized similarly, albeit less clearly for the N-terminally tagged version. It is known from our published work that the filaments labeled by $XopL_m$ are microtubules (MTs) [18]. As expected, the GFP control was localized to the cytosol and nucleus (Fig 2H).

While at first glance it appeared that wild-type $XopL_{Xe}$ and $XopL_m$ proteins differed in their localization pattern, upon closer examination a few MTs were also visibly labeled by $XopL_{Xe}$ in some cells (Fig 2C–2E, white arrows). It should be noted that in agroinfection assays epidermal cells are a mosaic of high expressing and lower expressing cells. In cells with low $XopL_{Xe}$ expression MT-labelling by $XopL_{Xe}$ was even more apparent (Fig 2D). This experiment told us three things: (i) wild-type $XopL_{Xe}$ has a similar localization pattern to $XopL_m$, (ii) the reduction in visible MT labeling in cells with high $XopL_{Xe}$ expression suggests that $XopL_{Xe}$ may disrupt MTs and (iii) abundant labeling of MTs in cells expressing $XopL_m$ compared with $XopL_{Xe}$ suggests that MT disruption occurs in an E3 ligase activity-dependent manner.

To confirm the identity of XopL-labeled filaments and to determine the fate of MTs bound by $XopL_{Xe}$, XopL-mCherry fusions were expressed in leaves of an *N. benthamiana* transgenic

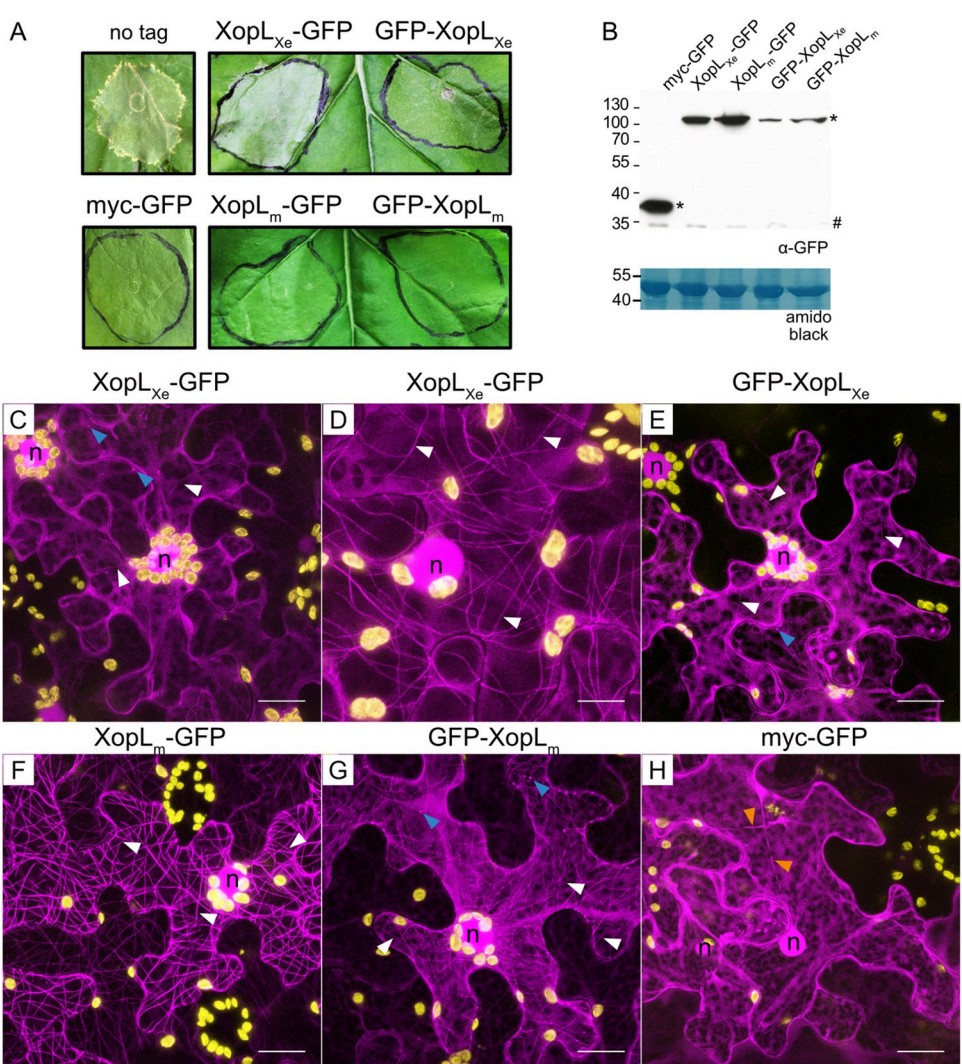

**Fig 2. XopL$_{Xe}$-induced cell death and MT-localization are negatively affected by an N-terminal fluorescence tag.** (A) Analysis of cell death induction following agroinfection of *N. benthamiana* leaves to express untagged XopL$_{Xe}$ and XopL$_m$ (E3 ligase-inactive, H584A, L585A, G586E) N- or C-terminally fused to GFP and GFP alone (OD$_{600}$ of 0.4). Plant reactions were monitored 5 dpi. (B) Western blot analysis of protein extracts isolated 2 dpi from the experiment depicted in (A). Protein signals were detected with a GFP-specific antibody (*). The left side of the blot shows protein mass in kDa; # indicates a non-specific signal. Amido black staining is included as a loading control. (C-G) Confocal microscopy pictures of lower epidermal cells 2 dpi from the experiment depicted in (A). Images depict the subcellular phenotypes of (C) XopL$_{Xe}$-GFP, (D) a lower expressing cell from the same inoculation spot as in (C) (imaging with the LSM 780 required a laser intensity of 16.8% laser power vs. 10% used in (C), with identical gain settings), (E) GFP-XopL$_{Xe}$, (F) XopL$_m$-GFP, (G) GFP-XopL$_m$ and (H) myc-GFP. GFP-labeled protein fusions are visible in magenta, plastid autofluorescence in yellow. Nuclei are labeled with 'n', blue arrows label mobile dots, white arrows point out microtubules and orange arrows label cytosolic strands. Scale bars are 20 μm for all images except (D), where the scale is 10 μm.

line with labeled MTs (GFP-TUA6; TUBULIN ALPHA-6 from *Arabidopsis thaliana* [21]). C-terminal XopL fusions were chosen for further localization studies since both C and N-terminal fusions exhibited similar sub-cellular phenotypes (i.e., plastid clustering) and localization patterns. In addition, a reduction in cell death compared to untagged XopL$_{Xe}$ and weaker MT binding of N-terminal fusions suggested that an N-terminal tag interferes with XopL activity. XopL$_{Xe}$ as well as the XopL$_m$ expression in tubulin-labeled plants confirmed the co-localization of these proteins with MTs (Fig 3), which does not occur when cells express the mCherry

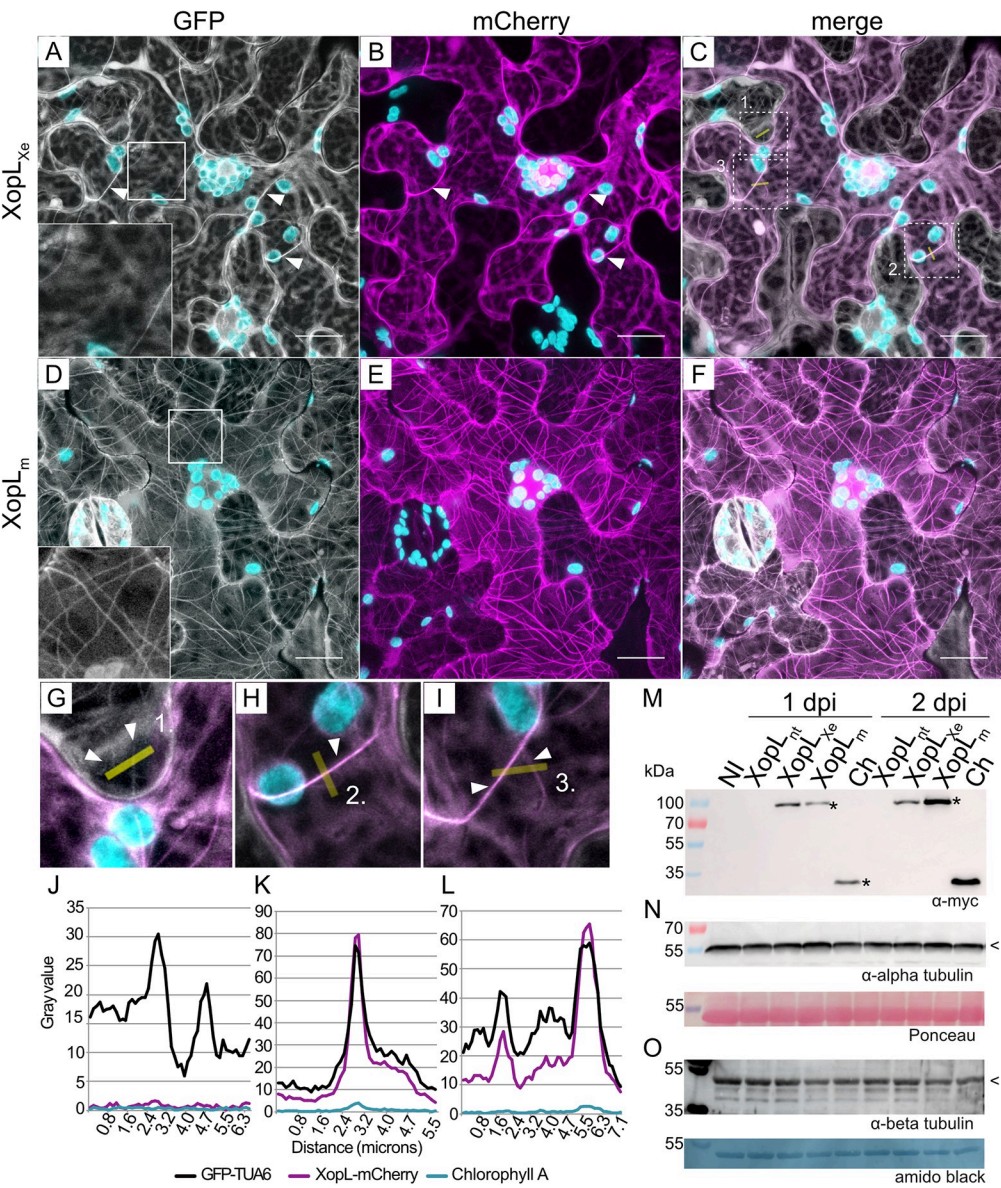

**Fig 3. XopL$_{Xe}$ destabilizes MTs but does not degrade tubulin.** (A-I) Confocal microscopy pictures of lower epidermal cells of GFP-TUA6 (labels MTs) transgenic *N. benthamiana* leaves. Leaves were agroinfected (OD$_{600}$ of 0.4) to express (A-C) XopL$_{Xe}$-mCherry and (D-F) XopL$_{m}$ -mCherry. Samples were harvested at 2 dpi for microscopy. The GFP channel is visible in white (MTs), the mCherry channel in magenta (XopL derivatives). (C) and (F) are merged GFP and mCherry channels. Plastids are visible in cyan, examples of MTs are labeled with white arrows. Scale bars are 20 μm. Magnifications from (C) are shown in (G-I); yellow lines represent the location of intensity plot measurements. (J-L) Intensity plots. (J) Shows a plot of a neighboring cell without XopL$_{Xe}$ expression (negative control). (K-L) Measurements across filaments in a XopL$_{Xe}$-expressing cell show co-localization. The signal from the chlorophyll channel was included as a negative control (cyan lines). (M-O) Western blot analysis of protein extracts isolated from *N. benthamiana* tissue that was not inoculated (NI) or agroinfected (OD$_{600}$ of 0.4) to express the myc-mCherry control (Ch) or XopL derivatives (XopL$_{nt}$ = untagged, XopL$_{Xe}$ and XopL$_{m}$ with C-terminal myc tags) 1 and 2 dpi. (M) Expression of proteins fusion was confirmed via detection with a myc-specific antibody. Expected size of myc-tagged proteins is marked with '*'. (N and O) The abundance of tubulin monomers was tested using antibodies specific to endogenous α and β tubulin. Expected tubulin size is marked with '<'. The left side of the blot shows protein mass in kDa. Ponceau and amido black staining of tubulin blots were used as loading controls.

protein alone (S2 Fig). In addition, as suspected, wild-type XopL$_{Xe}$ clearly reduced MT number compared to XopL$_m$ (Fig 3A compared to 3D) and the mCherry control (S2A Fig). Intensity plots across the few remaining MTs demonstrate the overlap of the XopL$_{Xe}$-mCherry and GFP-TUA6 signals (Fig 3K and 3L), which further confirms that XopL co-localizes with MTs. In contrast, control intensity plots in cells that did not express XopL (Fig 3J) or that expressed mCherry (S2G Fig), did not show co-localization. Together with data shown in Fig 2, this confirms that XopL$_{Xe}$ both localizes to MTs and triggers an E3 ligase-dependent decrease in MT abundance after agroinfection. These findings provide a plausible explanation as to why XopL$_{Xe}$ localization appeared to be limited to nucleus, cytosol, and dots in previous experiments [13, 18]. The decrease in MT number is not likely due to a decrease in tubulin subunits because α and β-tubulin levels were stable at 1 and 2 dpi; this time point is well within the timeframe of MT destruction (Fig 3M–3O).

## Taxol partially rescues MTs from XopL$_{Xe}$

To determine whether MTs could be rescued from XopL$_{Xe}$-mediated depolymerization we treated agroinfected tissues with the MT stabilizing agent taxol. Taxol reversibly binds to MTs via β-tubulin, altering its conformation to increase the rigidity and stability of filaments, independent of GTP binding, which is normally required for polymer growth [22]. Taxol treatments (dissolved in DMSO, see Materials and Methods) were administered at very early time points (4 h) following agroinfection, and microscopy was performed 2 dpi, when XopL$_{Xe}$ expression typically has eliminated most MTs. The result of taxol treatments on XopL$_{Xe}$-inoculated tissues was dramatic. MTs were indeed partially rescued and those that remained were highly bundled and unorganized, often forming loops (S3A Fig), all clearly labeled by XopL$_{Xe}$-mCherry (S3B and S3C Fig). DMSO controls at the same time point showed almost no MTs, as expected (S4A–S4C Fig). Treatment of the XopL$_m$ and mCherry controls with taxol clearly showed an increase in MT density and in the number of parallel filaments relative to the control (XopL$_m$ taxol treatment in S3D Fig compared to DMSO in S4D Fig; mCherry control treated with taxol in S3K Fig compared to DMSO in S4G Fig) illustrating the effectiveness of the taxol treatment. Taxol treatments of XopL$_{Xe}$ expressing cells resulted in a distinct 'spaghetti-like' conformation of MTs when compared to XopL$_m$ (compare S3A and S3D Fig). Taken together, taxol treatments not only provide further evidence for MT association by wild-type XopL$_{Xe}$, but differences in MT appearance between XopL$_{Xe}$- and XopL$_m$-expressing tissues treated with taxol also suggests that there are E3 ligase-dependent alterations to MT dynamics and/or structure that are not completely overcome by taxol.

## MT localization is not conserved between XopL$_{Xe}$ and XopL$_{Xcc}$

Having confirmed MT binding of XopL$_{Xe}$ *in planta* using multiple constructs, plant genotypes and taxol treatments, the next step was to determine if the same pattern could be observed in tissues expressing XopL$_{Xcc}$. For this purpose, translational XopL$_{Xcc}$-mCherry fusions were expressed in the GFP-TUA6 plants, allowing the simultaneous examination of protein localization and MT abundance. Like XopL$_{Xe}$, XopL$_{Xcc}$ was localized to the nucleus and the cytosol (Fig 4B), however, in contrast to XopL$_{Xe}$, it did not label MTs, nor did it trigger MT depolymerization (Fig 4A–4E). In addition, XopL$_{Xcc}$ did not induce plastid clustering as was seen with XopL$_{Xe}$ (Fig 4B compared to Figs 2C and 3B).

## MT localization is conserved among XopLs from *Xoo* and *Xac*

The inability of XopL$_{Xcc}$ to bind and trigger disassembly of MTs raised the question of whether MT-association/destabilization is a conserved characteristic of XopLs from other

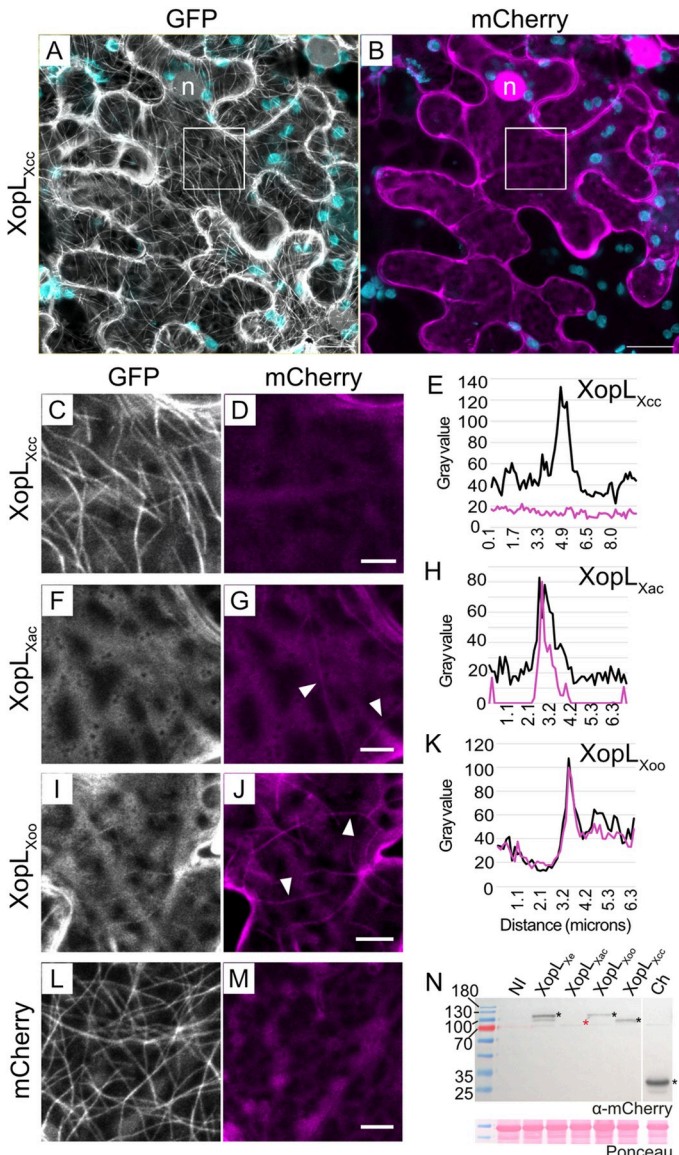

**Fig 4. XopL_{Xcc} does not localize to MTs.** Confocal microscopy pictures of lower epidermal cells of GFP-TUA6(labels MTs) transgenic *N. benthamiana* leaves. Leaves were agroinfected (OD_{600} of 0.4) to express XopL proteins translationally fused to a C-terminal mCherry. Samples were harvested 2 dpi for microscopy. The GFP channel is visible in white (MTs), the mCherry channel in magenta (XopL derivatives) and plastids in cyan. (A-B) Plant cell expressing XopL_{Xcc}-mCherry. Scale bars are 20 μm. Insets from (A) and (B) (area outlined with a white box) are magnified in (C) and (D), respectively. (F-G) show a cell expressing XopL_{Xac}-mCherry and (I-J) XopL_{Xoo}-mCherry. (L-M) is the mCherry control. Scale bars for (C-M) are 5 μm. Examples of MT labeling are indicated by white arrows. (E, H and J) Fluorescence intensity plot across MTs to test for co-localization of GFP-TUA6 with (E) XopL_{Xcc}, (H) XopL_{Xac} and (K) XopL_{Xoo}. Magenta lines show the intensity of the XopL-mCherry signal, black represents GFP-TUA6. (N) Western blot analysis of protein extracts isolated 2 dpi from tissue used for microscopy in (A-M). Signals were detected using an mCherry-specific antibody (*), with exception of XopL_{Xac} (*) which was not detectable. Samples that were not inoculated 'NI' or expressing mCherry 'Ch' were loaded as controls. The left side of the blot shows protein mass in kDa. Ponceau staining shown as a loading control.

xanthomonads or specific to XopL from *Xe* 85–10. Therefore, codon-optimized XopLs from *Xoo* and *Xac* were fused in frame with a C-terminal mCherry for localization studies in GFP-TUA6 stable transgenic *N. benthamiana* plants. At 2 dpi, MTs were already degraded in the presence of XopL_{Xac} and XopL_{Xoo}, but not in the presence of mCherry (Fig 4F–4M). In

some cases, a few MTs remained and were labeled by $XopL_{Xac}$ and $XopL_{Xoo}$, demonstrating again that they also localize to MTs (white arrows, Fig 4G and 4J and intensity plots in Fig 4H and 4K). As previously reported, $XopL_{Xoo}$ was also cytosolic [17] and appeared to be nuclear-excluded (S5B Fig), whereas $XopL_{Xac}$ was cytosolic and nuclear (S5A Fig). This means that, while nuclear localization varied, MT binding and destabilization is a conserved feature of several XopLs originating from *Xanthomonas* species closely related to *Xe*, but not the more distantly related *Xcc*. A simple explanation for why previous reports on $XopL_{Xoo}$ [17] and $XopL_{Xe}$ [13] overlooked MT localization is that MTs were likely dismantled at the time points chosen after agroinfection (2–3 dpi). The conservation of MT localization among XopLs from multiple *Xanthomonas* species suggests the potential functional importance of this characteristic. In contrast, but equally exciting is the finding that the more distantly related $XopL_{Xcc}$ does not exhibit MT localization, thus providing a unique chance to evaluate how the differences in localization may have evolved and how it affects effector function.

Unfortunately, codon optimization of XopLs for *N. benthamiana* resulted in very poor expression levels (Fig 4N). While proteins were sufficiently expressed for microscopy, expression was patchy and low with little signs of macroscopic cell death (S5C Fig), even in the case of $XopL_{Xe}$ and $XopL_{Xoo}$, where strong cell death has been well documented in agroinfected *N. benthamiana* [12, 17]. For this reason, codon-optimized XopLs were not used to evaluate cell death phenotypes going forward.

### The XL-box from $XopL_{Xe}$ is not required for MT binding

As mentioned, $XopL_{Xe}$ is composed of an α-helical region (3 alpha helices; herein referred to as 'α'), LRR domain (LRR) and XL-box (XL) which have been crystalized, as well as an unstructured N-terminus (NT), which harbors the type III translocation signal [12]. To get a rough idea of which domains are important for the MT binding, XopL was divided into two parts, as was done by Singer et al. [12]. Amino acids 1–450 (referred to as NTαLRR) and 451–660 (XL) were separately fused to GFP and expressed in *N. benthamiana* via agroinfection (constructs outlined in Fig 5A). Anticipating that, if the XL-box was responsible for MT binding, it would also have the capacity to degrade filaments, we also included an E3 ligase-inactive version (451–660 aa, H584A_L585A_G586E; $XL_m$). The result was clear: NTαLRR-GFP was sufficient for MT association, whereas the XL-box was not required (compare Fig 5C to 5F and 5G). XopL NTαLRR was further divided into two parts, the NT (1–136 aa) and the αLRR (137–450 aa). Since the LRR domain is generally believed to be responsible for protein substrate binding, it was surprising that very little binding was observed for the αLRR (Fig 5E) and αLRR_$XL_m$ (Fig 5H) derivatives. There was also no MT association in the case of NT expression (Fig 5D). Only the combination of NT plus the αLRR was sufficient to confer MT binding comparable to the full-length $XopL_m$ protein (Fig 1E). All proteins were detectable at the correct size in Western blots (Fig 5L).

### Comparative analysis of XopLs identified prolines contributing to MT binding stability of $XopL_{Xe}$

Differences in protein localization of $XopL_{Xcc}$ and $XopL_{Xe}$ prompted a comparison of amino acid sequences to narrow down the region of $XopL_{Xe}$ that confers MT association. $XopL_{Xcc}$ and $XopL_{Xe}$ are 46.9% similar and 34.2% identical at the amino acid level, with most similarity lying within the LRR (70.5% similarity, 53.7% identity), followed by the XL-box (48.5% similarity, 32.7% identity) and then the α region (45.2% similarity, 29% identical). The LRR domain was previously predicted to be responsible for protein substrate recognition [12], and it was initially suspected it may be important for interaction with MTs, as opposed to the XL-

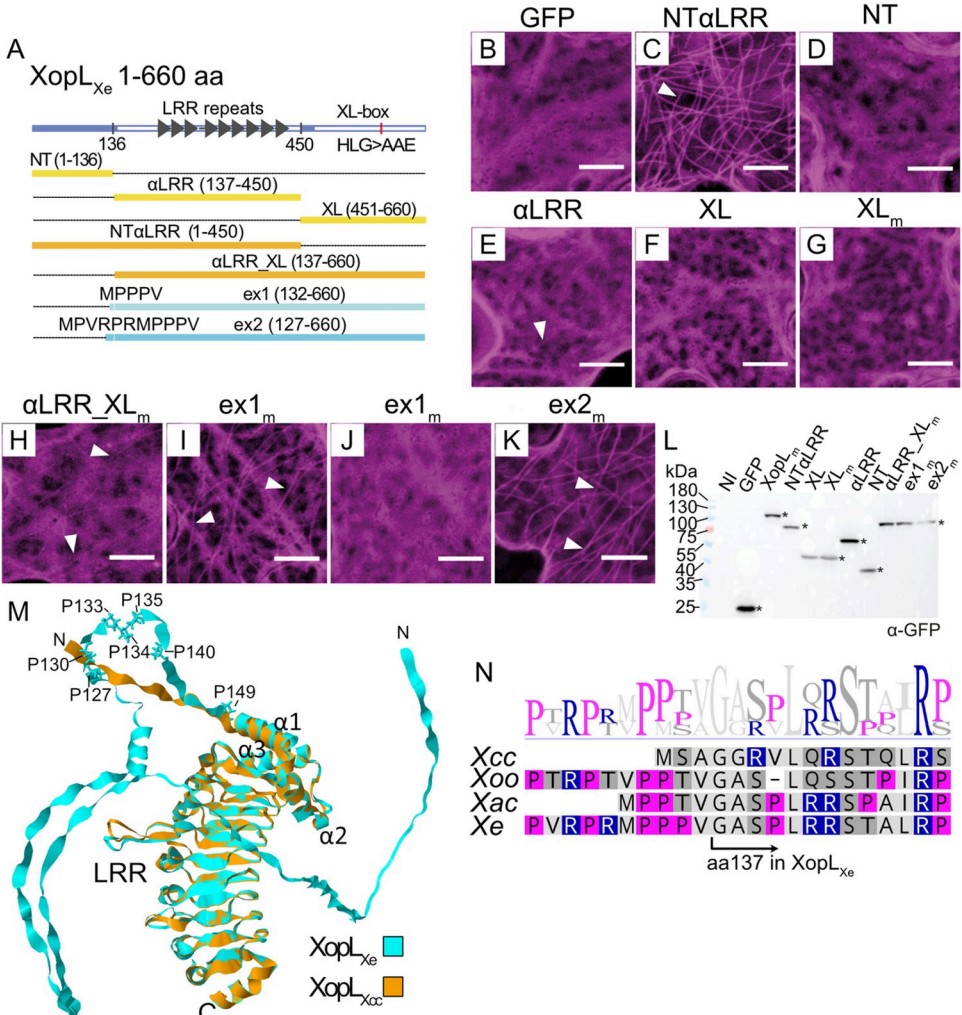

**Fig 5. Comparative sequence analysis of XopL homologs identifies a proline-rich-region (PRR) important for MT localization.** (A) Domain structure of XopL$_{Xe}$ and derivatives used to functionally analyze the PRR. Domains crystalized by Singer et al. (2013) are denoted with a white bar, LRR repeats by dark gray arrows. Amino acid exchanges in the XL-box of XopL$_{Xe}$ to generate XopL$_m$ are indicated. Single domain derivatives are annotated in yellow, multiple domain in orange, and 'extension' (ex) constructs in blue. Amino acids added to the αLRR_XL in ex1 and ex2 are shown at the N-terminus of the corresponding construct. Amino acid positions included in each derivative are shown in brackets. (B-K) Confocal microscopy pictures of lower epidermal cells of wild-type *N. benthamiana* leaves. Leaves were agroinfected (OD$_{600}$ of 0.4) to express (B) a GFP control, (C) NTαLRR, (D) NT, (E) αLRR, (F) XL, (G) XL$_m$, (H) αLRR_XL$_m$, (I) ex1$_m$; cell with MT localization, (J) ex1$_m$; cell without MT localization and (K) ex2$_m$. Subscript 'm' indicates a triple amino acid exchange in the E3 ligase domain (H584A, L585A, G586E) that renders the derivative E3 ligase-inactive. Samples were harvested at 2 dpi for microscopy. All XopL derivatives are tagged at the C-terminus with GFP (visible in magenta). Examples of MT association are indicated by white arrows. Scale bars are 5 μm. (L) Western blot analysis of protein extracts isolated 2 dpi from experiment in (B-K). Signals were detected using a GFP-specific antibody (*). The left side of the blot shows protein mass in kDa. (M) An alignment of XopL$_{Xcc}$ (orange) and XopL$_{Xe}$ (cyan) 3D structural models generated in AlphaFold2. The alignment includes the unstructured NT, α-helical (α1–3) and LRR regions. The cluster of prolines conserved in MT-localizing XopLs are labeled and represented in stick format. N- and C-termini are marked with N and C, respectively. Models do not include the C-terminal XL-box. (N) Alignment of XopL$_{Xcc}$, XopL$_{Xe}$, XopL$_{Xac}$ and XopL$_{Xoo}$ at the junction between NT and αLRR constructs. Amino acids are colored based on polarity (Geneious Prime), basic amino acids are in blue, and fuchsia highlights prolines. The boundary between the NT and αLRR constructs is annotated at amino acid 137. The sequence logo above the alignment shows sequence conservation at specific positions.

box. While indeed the XL-box is not required, the finding that the αLRR was insufficient for $XopL_{Xe}$ localization to MTs caused us to take a closer look at the NT sequences of these two proteins. NTs are the most variable region among XopLs and vary greatly in length between species [12]. AlphaFold2 structural predictions of the $XopL_{Xe}$ and $XopL_{Xcc}$ proteins resulted in very similar 3D models within the αLRR and highlighted the presence of a long unstructured NT in $XopL_{Xe}$ that is absent from $XopL_{Xcc}$ (Fig 5M). However, the shorter NT of $XopL_{Xcc}$ is not sufficient to explain the lack of MT binding, since that of $XopL_{Xac}$ is even shorter and is able to bind and destroy the MT network in a manner comparable to XopLs with extended NTs ($XopL_{Xe}$ and $XopL_{Xoo}$; see alignment of all 4 XopLs in S6 Fig).

To identify potential sequence differences within the NT that contribute to MT association, the sequences of the three MT-binding XopLs were compared with the non-MT localized $XopL_{Xcc}$. Closer examination of the boundary between the NT and αLRR region revealed that the N-terminal boundary chosen for the αLRR construct (position 137) excluded 5 amino acids (MPPPV) that were highly similar in sequence to that of the short NT of $XopL_{Xac}$ (MPPTV) (Fig 5M). Interestingly, two of these amino acids are prolines which were 100% conserved in MT-associated XopLs but are conspicuously absent from $XopL_{Xcc}$. In addition, these amino acids are part of a larger proline-rich region (PRR) present in all MT-binding XopLs (Fig 5N).

Suspecting that the PRR may facilitate XopL MT binding, the $αLRR\_XL_m$-GFP fusion was extended N-terminally to include the MPPPV motif (132–660 aa), henceforth referred to as extension 1 ($ex1_m$). Additionally, a second construct including the entire PRR ($MPVRPRMPPPVαLRR\_XL_m$, 127–660 aa) was generated ($ex2_m$). Excitingly, $ex1_m$ reconstituted MT binding when compared to the $αLRR\_XL_m$, and some cells showed clear MT localization (Fig 5I). However, MT localization was not consistent, and some cells showed nearly no labeling (Fig 5J). In contrast, $ex2_m$ reliably reconstituted binding, resembling $XopL_m$ localization (Fig 5K). This means, that with the addition of only 11 amino acids the $αLRR\_XL_m$ was transformed from a non-binding to a MT-binding derivative. The discovery of a conserved PRR in XopL is particularly exciting since PRRs are known to stabilize direct interactions with MTs in both animal [23] and plant MT-associated proteins [24].

## $XopL_{Xe}$ binds MTs *in vitro*

The fact that $XopL_{Xe}$ localizes to MTs *in planta*, combined with the presence of a conserved PRR specific to XopLs that associate with MTs led to the question of whether this localization pattern is the result of a direct interaction with MTs, or rather an association with an MT-bound protein. To test for direct MT binding, co-sedimentation assays were performed. In this assay, MTs are polymerized *in vitro*, the protein of interest is incubated with MTs, and the mixture is subjected to ultracentrifugation to pellet MTs together with bound proteins. Thus, binding to MTs is indicated by the enrichment of the protein of interest in the pellet fraction. It should be noted that the protein of interest must be soluble in low salt and room temperature since these conditions are optimal for tubulin polymer stability (see Materials and Methods). In the case of $XopL_{Xe}$ this provided a challenge, as $XopL_{Xe}$ was not soluble in low salt conditions in our hands. However, solubility was optimized by expressing a fusion of $XopL_{Xe}$ to 6xHis-StrepII-SUMO, pH optimization and the addition of aminocaproic acid (see S7A Fig for details).

The co-sedimentation assay revealed that XopL is present in the supernatant both when MTs are absent and present, albeit the XopL signal in the supernatant is slightly reduced when MTs are included (Fig 6A left panel). More exciting, however, was that XopL was enriched in the pellet fraction in the presence of MTs when compared to samples without MTs (Fig 6A

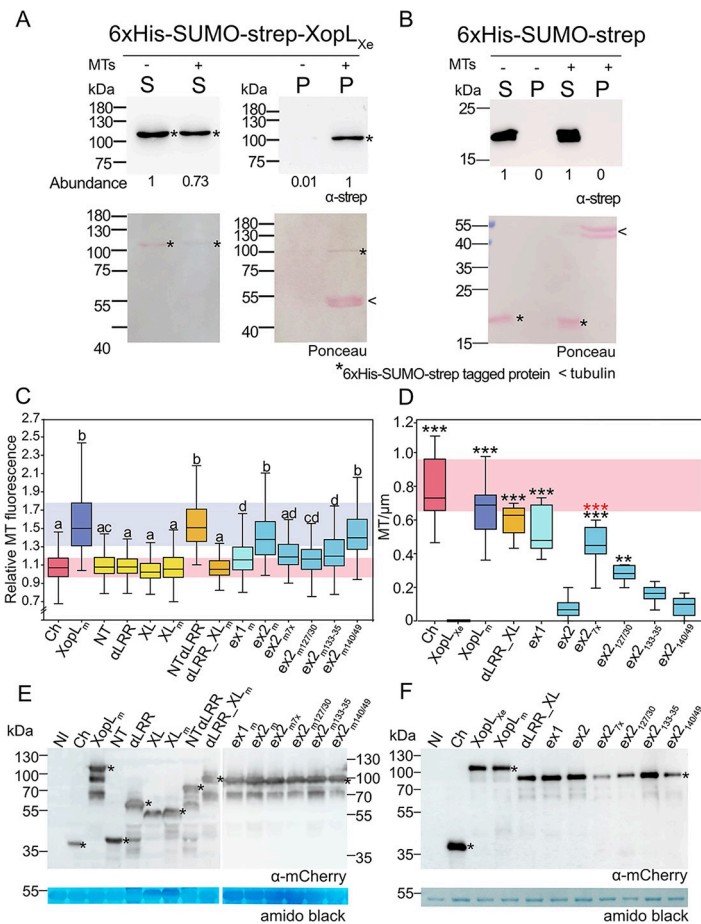

**Fig 6. Prolines within the PRR are essential to MT-association of XopL_Xe.** (A-B) Western blot analysis of MT co-sedimentation assay utilizing (A) 5 µg of affinity-purified XopL_Xe (88.1 kDa) protein tagged with 6xHis-SUMO-strep or (B) the tag-alone control. Supernatant fractions are marked 'S' and pellet fractions 'P'. Samples with MTs are marked '+' and without by '-'. Recombinant protein was detected using a strep-specific antibody (*). The intensity of the XopL signal (abundance) is shown below corresponding lane. Ponceau stained membranes show tubulin at 55 kDa in the pellet fraction with MTs (<). The left side of the blots show protein mass in kDa. Experiments with XopL were repeated at least three times and the tag-only control experiment was repeated two times. (C) Quantification of MT-association of XopL_m derivatives from confocal images of agroinfected GFP-TUA6 lower leaf epidermis (2 dpi, $OD_{600}$ of 0.8). MT binding is expressed as 'Relative MT fluorescence' (see Materials and Methods). Treatments that are not significantly different (p>0.05) are labeled with the same letter (Kruskal-Wallis One Way Analysis of Variance on Ranks, pairwise Dunn's post-hoc). Background colors are included for ease of comparison between key controls XopL_m (blue) and mCherry (red). (D) Quantification of confocal images to evaluate MT density following the expression of XopL_Xe derivatives in agroinfected GFP-TUA6 lower leaf epidermis (2 dpi, $OD_{600}$ of 0.8). Treatments that were significantly different than XopL_Xe are marked with black asterisks (* = p<0.05, ** = p<0.01, *** = p>0.001; Kruskal-Wallis One Way ANOVA on Ranks, Dunn's post-hoc against XopL). ex2 proline to alanine exchanges that were significantly different than ex2 are marked with red asterisks (*** = p>0.001; Kruskal-Wallis One Way ANOVA on Ranks, Dunn's post-hoc against ex2). Red box in the background is included for ease of comparison with the mCherry control (red). (C and D) Boxes represents first to third quartiles and the median is represented by a horizontal line. Whiskers represent the distribution of remaining data points. (E and F) Western blot analysis of protein extracts isolated from the experiments in (C) and (D), respectively. Signals of XopL derivatives and the mCherry control were detected using an mCherry-specific antibody (*). 'NI' and 'Ch' refer to not inoculated and mCherry control, respectively. Protein mass is expressed in kDa. (F) Amido black staining is shown as a loading control.

right panel); a strong indication for direct MT binding. As a control to ensure that MTs were indeed polymerized western blots were Ponceau stained. This showed that tubulin was only detectable in the pellet fraction (Fig 6A, lower right panel). Even in the ponceau staining XopL enrichment in the pellet fraction was visible. It should be noted that in some cases we did detect some XopL in the pellet fraction in the absence of MTs, especially in cases where more protein was used (S7B Fig). This was not surprising, given the low solubility of XopL at low salt and room temperature. However, even in these experiments, XopL was clearly enriched in the pellet fraction when MTs were present. Given that the MT co-sedimentation assay suggests that XopL binds directly MTs, the implications of a PRR that contributes to binding warranted further investigation.

## Quantification of MT association of $XopL_{Xe}$ variants confirms the importance of prolines to MT association

To allow for the quantification of MT localization, XopL E3 ligase mutant derivatives described above were fused to mCherry and expressed in GFP-TUA6 plants. A method for quantification was devised using the open-source image analysis platform, Fiji [25, 26]. 'Relative MT fluorescence' was quantified by first measuring mCherry signals at MT and dividing by measurements taken in neighboring cytosol (for details, see Materials and Methods). It is worth noting that in cases where the background is high, such as $\alpha LRR\text{-}XL_m$, weak MT labeling was observed in some cells but this was not reflected in measurements. Despite this, however, quantification largely reiterates what is seen in Fig 5. Individual $XopL_{Xe}$ domains and $\alpha LRR\text{-}XL_m$ do not co-localize with TUA6-labeled MTs and are comparable to labeling by an mCherry control, whereas $XopL_m$ and NT$\alpha$LRR fluorescence signals were enriched along MTs when compared to the background (Fig 6C). $Ex1_m$ binding was variable, but significantly enhanced compared to the mCherry control, and $ex2_m$ enhanced binding further.

This approach was also used to evaluate the importance of the 7 conserved prolines in the PRR. Four $ex2_m$ derivatives were generated with proline to alanine exchanges: P127A + P130A ($ex2_{m127/30}$), P133-35A ($ex2_{m133\text{-}35}$), P140A + P149A ($ex2_{m140/49}$) and a variant with all 7 exchanged ($ex2_{m7x}$). Expression of mCherry-tagged variants in GFP-TUA6 plants revealed that $ex2_{m127/30}$ and $ex2_{7x}$ had the greatest decrease in MT binding, followed by $ex2_{m133\text{-}35}$, whereas $ex2_{m140/49}$ did not have a significant impact (Fig 6C). Protein sizes and stability were confirmed via Western blot (Fig 6E). In conclusion, the quantification and analysis of the MT association of proline exchange mutants further confirms that the PRR identified through sequence comparison of $XopL_{Xe}$ and $XopL_{Xcc}$ contributes to MT binding of $XopL_{Xe}$.

## $XopL_{Xe}$-induced decrease in MT abundance is correlated with MT affinity

The above-mentioned test for MT-association of PRR truncations and alanine exchanges was conducted using E3 ligase-inactive variants, to avoid MT disassembly. To determine if the level of MT-association of these variants was sufficient to destroy MTs, new mCherry fusions were generated with active E3 ligase domains. These new fusions were expressed in GFP-TUA6 plants to monitor MT abundance. Quantification of MT number revealed that indeed the addition of the proline rich region (127–136 aa) onto the $\alpha LRR\_XL$ was sufficient, not only to bind MTs, but to decrease MT number very close to the level of the WT $XopL_{Xe}$ (Fig 6D). This experiment also shows that exchange mutants $ex2_{7x}$ and $ex2_{127/30}$ resulted in significant MT rescue compared to $XopL_{Xe}$, whereas $ex2$ and $ex2_{133\text{-}35}$ and $ex2_{140\text{-}49}$ were not significantly different from wild-type $XopL_{Xe}$.

When relative MT fluorescence exhibited by $XopL_m$ derivatives was plotted against the MT number following expression of the E3 ligase-active versions it revealed a negative correlation,

i.e., the more binding a derivative exhibited, the less MTs remained (S8 Fig). This data not only suggests that the strength of XopL MT binding is correlated to the loss of MTs, but also emphasizes the contributions of prolines 127/30 to this activity.

## NTα of XopL$_{Xe}$ converts XopL$_{Xcc}$ into a MT binding protein

The unstructured NT of XopL$_{Xe}$ alone was not sufficient for MT localization in the experiments described in Fig 5, but this construct lacked part of the PRR region. Therefore, to test if the NT+PRR is sufficient for MT binding, the NT was extended C-terminally to include amino acids 1–149 (NT$_{149}$) and tagged with mCherry. However, this construct is insufficient for MT localization (Fig 7C), suggesting that structural components are missing that facilitate an interaction with MTs. This fits with data on PRRs in well characterized MT binding proteins from humans, such as Tau [27] and MAP2 [28], where PRRs are required to stabilize protein-MT associations but are not sufficient for binding alone. Interactions with MTs often require stretches of surface exposed basic amino acids to facilitate electrostatic interactions with acidic residues in tubulin [29]. Therefore, we re-examined the amino acid sequence of XopL$_{Xe}$ downstream of position 149; the α region (made up of 3 helices) is enriched in basic amino acid residues when compared to XopL$_{Xcc}$, a feature that is notably conserved among MT-associated XopLs (S6 Fig). Electrostatic surface potential models of XopL$_{Xe}$ and XopL$_{Xcc}$ proteins predict visible structural differences in the surface of the α region (Fig 7G). Residues from XopL$_{Xe}$ α2 and α3 are predicted to form a surface exposed, positively charged region that is absent in XopL$_{Xcc}$. We tested the importance of the α region for interactions with MTs by sequentially adding α-helices 1–3 to the 1–149 aa derivative of XopL to create constructs 1–164 aa (NTα1)-, 1–179 aa (NTα1–2)- and 1–210 aa (NTα)-mCherry fusions. Analysis of these constructs in GFP-TUA6 plants revealed very weak binding of NTα1 (Fig 7D), whereas extensions towards the C-terminus as in NTα1–2 and NTα incrementally increased in their affinity for MTs (Fig 7E and 7F). Quantification of MT-association of the NTα variants further confirms this finding, with only the NTα construct demonstrating MT-association approaching that of XopL$_m$ (Fig 7K, p>0.05). These results suggest significant contributions of the α-helices of XopL$_{Xe}$ in establishing a stable interaction with MTs.

The finding that the NTα was sufficient for MT localization of XopL$_{Xe}$ guided a domain swap experiment between XopL$_{Xe}$ and XopL$_{Xcc}$ to create a version of XopL$_{Xcc}$ that can bind MTs. Hence, the NTα of XopL$_{Xe}$ (1-210aa) was added to the LRR of XopL$_{Xcc}$ (78–428) and fused with a C-terminal mCherry to produce the E3 ligase-inactive construct NTα$_{Xe}$LRR$_{Xcc}$. NTα$_{Xe}$LRR$_{Xcc}$ clearly localized to MTs (Fig 7H–7J), re-iterating that the NTα and not the LRR or XL-box are needed for MT association.

## XopL MT binding is correlated with cell death

As mentioned, one readout of XopL$_{Xe}$ activity is that it induces cell death when expressed in *N. benthamiana* by agroinfection, a phenotype that was previously shown to rely on both the LRR and the XL-box together, likely due to the ubiquitination of an unknown plant protein [12]. The question arises, is the cell death phenotype linked to the decrease in MTs? To test this, we used the XopL$_{Xe}$ E3 ligase-active truncations and amino acid exchange mutants to test for cell death induction. Cell death was monitored via red fluorescence scanning as described in detail by Xi et al. [30]. For these experiments, all derivatives were fused with a C-terminal GFP tag so that their expression would not interfere with detection of cell death in the mCherry channel. As previously published [12], XopL$_{Xe}$ induced the strongest cell death, indicated by high gray values, whereas the GFP control showed almost no fluorescence (Fig 8A). Visual inspection of the leaves confirmed this (Fig 8C). In addition, constructs that strongly

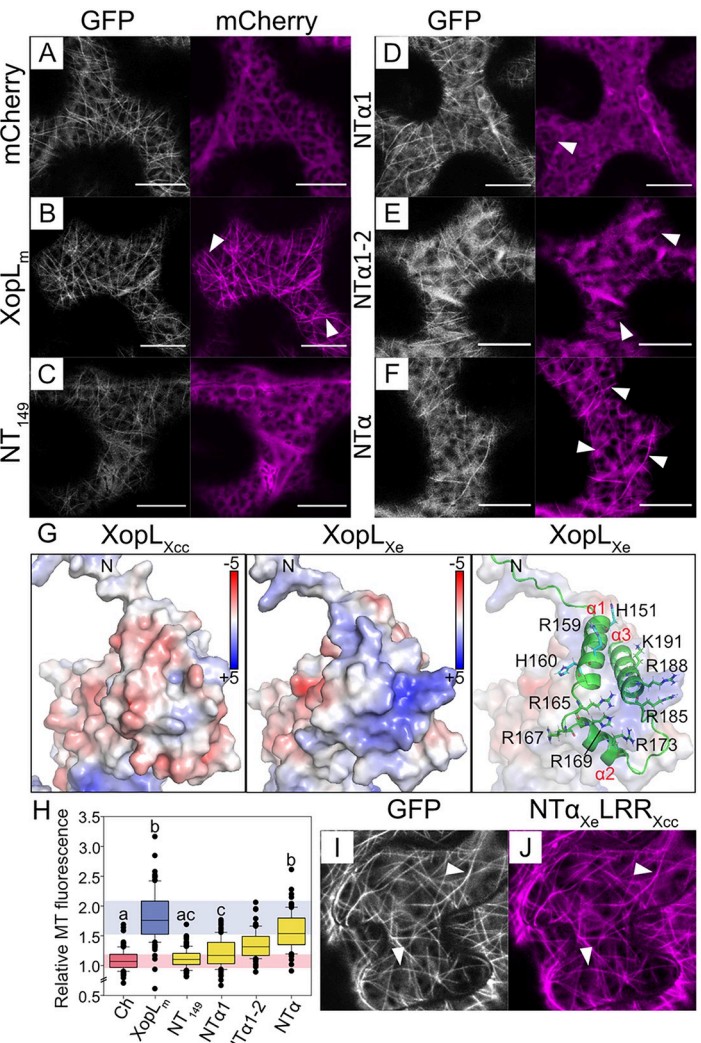

**Fig 7. The exchange of the NTα region of XopL_Xe and XopL_Xcc is sufficient to engineer a MT-binding XopL_Xcc.**
(A-F, H-J) Confocal microscopy images of lower epidermal cells of GFP-TUA6 (labels MTs) transgenic *N. benthamiana* leaves. Leaves were agroinfected (OD_600 of 0.8) to express (A) mCherry and (B) XopL_m controls and (C-F) XopL derivatives: (C) XopL_Xe 1–149 aa (NT_149), (D) XopL_Xe 1–164 aa (NTα1), (E) XopL_Xe 1–179 aa (NTα1–2), (F) XopL_Xe 1–210 aa (NTα; α1–3). Scale bars are 15 μm. The mCherry channel is visible in magenta and GFP (MTs) is white. White arrows point out examples of MT labeling. The corresponding western blot is shown in S9 Fig. (G) Electrostatic surface potential comparison of the α-helical regions of XopL_Xcc and XopL_Xe modelled with AlphaFold2/ PyMOL. Values are expressed on a color scale from -5 (red) to +5 (blue) kT/e. The right panel depicts an overlay of the electrostatic map of XopL_Xe with its ribbon structure to show the locations of α 1–3. Basic residues are represented in stick format. N-terminal ends of proteins are marked 'N' in all panels (H) Quantification of MT-association of XopL derivatives from confocal images of agroinfected GFP-TUA6 lower leaf epidermis (2 dpi, OD_600 of 0.8) depicted in (A-F). MT binding is expressed as 'Relative MT fluorescence' (see Materials and Methods). Treatments that are not significantly different (p>0.05) are labeled with the same letter (Kruskal-Wallis One Way Analysis of Variance on Ranks, pairwise Tukey post-hoc). Background colors are included for ease of comparison between key controls XopL_m (blue) and mCherry (red). Boxes represents first to third quartiles and the median is represented by a horizontal line. Whiskers represent the distribution of remaining data points. (I-J) Confocal microscopy image of XopL_Xe 1–210 aa translationally fused with XopL_Xcc 78–428 aa (NTα_Xe LRR_Xcc) from the same experiment as depicted in (A-F). (I) GFP labeled MTs are in white, (J) the domain swapped XopL in magenta. White arrows point out examples of MT labeling. Scale bars are 15 μm. The corresponding western blot is shown in S9 Fig.

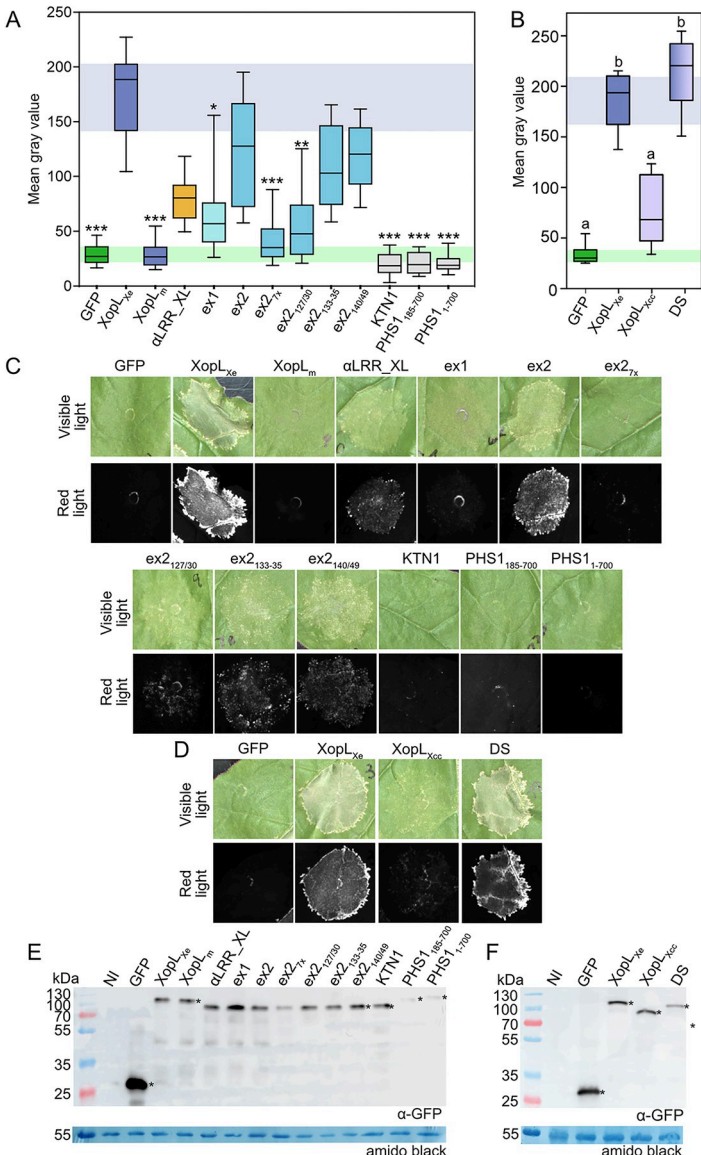

**Fig 8. Cell death is correlated with MT association of XopL.** (A-B) Quantification of plant cell death via red fluorescence scanning of agroinfected leaves expressing XopL derivatives tagged with GFP 5 dpi ($OD_{600}$ of 0.8). (A) Cell death in leaves expressing MT-binding and non-binding derivatives of $XopL_{Xe}$ or Arabidopsis proteins KTN1 and PHS1. Treatments that were significantly different than $XopL_{Xe}$ are marked with asterisks (* = $p < 0.05$, ** = $p < 0.01$, *** = $p > 0.001$; Kruskal-Wallis One Way ANOVA on Ranks, Dunn's post-hoc test against $XopL_{Xe}$). (B) Cell death following expression of $XopL_{Xe}$, $XopL_{Xcc}$, and $NT\alpha_{Xe}LRR_{Xcc}XL_{Xcc}$; termed DS. Treatments that are not significantly different are marked with the same letter ($p > 0.05$; One way analysis of variance, pairwise Bonferroni t-tests post-hoc). (A-B) Boxes represent first to third quartiles, while the median is marked by a horizontal line and whiskers show the distribution of remaining data points. (C-D) Examples of visible light and red-light photos of agroinfection spots recording cell death from the experiments in (A) and (B) respectively. (E) and (F) Western blot analysis of protein extracts isolated from tissue analyzed in experiments in (A) and (B), respectively. Signals were detected using a GFP-specific antibody (*). The left side of blots shows protein mass in kDa. Amido black staining was included as a loading control.

localized to MTs such as ex2, $ex2_{140/149}$ and to a lesser extent $ex2_{133-35}$, were most efficient at inducing cell death (Fig 8A). All derivatives were expressed at a similar level and were detectable at the expected size (Fig 8D). We therefore concluded that there was a strong link between MT-association of $XopL_{Xe}$ and cell death induction.

To explore the link between cell death and MT binding the next experiments aimed to quantitatively confirm data shown in Fig 1A which suggested that the non-binding XopL$_{Xcc}$ does not cause cell death. Furthermore, a domain swap was used to generate an E3 ligase-active MT-associated XopL$_{Xcc}$ variant (NTα$_{Xe}$LRR$_{Xcc}$XL$_{Xcc}$; termed DS for short). All proteins were fused to a C-terminal GFP to allow cell death measurements via red fluorescence scanning. Confirming results from Fig 1A, XopL$_{Xcc}$ showed little signs of cell death at 5 dpi when compared to GFP controls. In contrast, expression of the DS construct resulted in strong cell death reactions that were comparable to XopL$_{Xe}$ and significantly higher than with XopL$_{Xcc}$ (Fig 8B and 8C). Protein abundance of derivatives was similar (Fig 8E). Cell death at early time points (between 1–2 dpi) interfered with the ability to assess whether MTs were disassembled in these cells. These results strongly support the idea that XopL binding to MTs is linked to cell death, but also the idea that changing the subcellular localization of XopLs may be sufficient to alter their activity *in planta*.

## MT destruction is not sufficient to induce cell death

Due to the importance of MTs for cell structure and general function, it is possible that the cell death caused by XopL is simply due to the lack of MTs in XopL-expressing cells. To test this hypothesis, MTs were disrupted via the overexpression of known MT disrupting proteins from *A. thaliana*: the MT severing protein KATANIN 1 (KTN) [31] and truncated versions of the tyrosine phosphatase PHS1, a protein involved in cytoskeleton rearrangement during stress [32]. Two phosphatase-dead variants of PHS1 were used: aa 1–700 and aa 85–700 (aa 85–700 lacks the kinase interaction motif), both of which were previously shown to depolymerize MTs [32]. MT disruption by these proteins was confirmed via over-expression in GFP-TUA6 plants (S10A–S10H Fig). This approach was chosen in place of chemical treatments, such as oryzalin, which sequesters tubulin to prevent MT formation [33]. Oryzalin fails to eliminate all MTs in our experimental setup [18] and is not effective over the time-frame desired (1–2 dpi). Surprisingly, even after 5 dpi, no significant cell death was observed when either PHS or KTN1 was expressed (Fig 8A and 8C). MT destabilization by XopL, therefore, is unlikely the cause of cell death. To determine whether the cell death was simply an additive effect of XopL E3 ligase activity and MT destruction occurring in the same cells, the αLRR_XL variant, which does not bind MTs, was co-expressed with the KTN1 and PHS$_{185-700}$ constructs. Quantification of cell death showed that the expression of the non-binding αLRR_XL and ex2$_{127/30}$ in cells where MTs were destabilized by PHS1 or KTN1 was insufficient to induce cell death comparable with XopL$_{Xe}$ (S10 Fig). Taken together, the data suggest that MT-localized XopL$_{Xe}$ activity is important for this phenotype.

## Discussion

The T3E XopL is present in most sequenced *Xanthomonas* strains, and, thus, considered to be a conserved ancestral effector [7, 11]. XopLs are characterized by a conserved LRR region (implicated in protein-protein interactions), and the XL-box, which was identified as a novel protein fold conferring E3 ligase activity [12]. However, despite similarities in domain arrangement, protein structure, and E3 ligase activity, this new study demonstrates that there is considerable functional variation between XopL proteins originating from different *Xanthomonas* species. Comparative analyses of the subcellular localization of XopLs after agroinfection revealed that XopL from different *Xanthomonas* species (*Xe*, *Xac* and *Xoo*) associate with microtubules (MTs) *in planta*, ultimately triggering their breakdown. In the case of XopL$_{Xe}$ we were also able to show that MT-association is strongly correlated with plant cell death. This is in stark contrast to XopL from *Xcc*, which did not exhibit any of these phenotypes. Exploiting

the naturally occurring differences in MT-binding between XopL homologs, sequence comparisons identified a PRR and α-helical region important for MT-association of XopL$_{Xe}$. Manipulating or truncating the PRR sequence to alter MT-association demonstrated a strong link between MT-associated activity of XopL$_{Xe}$ and cell death induction. Thus, despite structural similarities not all XopLs exhibit the same activity, suggesting that differences in the subcellular localization of XopL proteins conferred during evolution may play an important role in their biochemical activities.

## Conservation of XopL PRR across *Xanthomonas* clades

*In vitro* MT co-sedimentation assays and *in planta* analyses of XopL truncations/amino acid substitutions suggest that the interaction of XopL with MTs is direct and not mediated by MT-associated proteins. XopL$_{Xe}$ association with MTs requires the unstructured N-terminus, including a PRR, in concert with an α-helical region. Sequence comparisons of XopL proteins from 24 different species scattered throughout the *Xanthomonas* genus revealed that, while the NT varies in length, the PRR is a conserved feature among XopLs (S11 Fig). In this context it is also notable that the PRR co-occurs with a high density of basic amino acids (e.g., 19.6% in XopL$_{Xe}$ vs. 9.9% in XopL$_{Xcc}$) within the α region. The conservation of the PRR and α-helical region in combination suggests that MT-localization is likely common to XopL proteins throughout the *Xanthomonas* genus. Interestingly, a strain of *X. hyacinthi* (CFBP1156) which shares a common ancestor with *Xe*, *Xoo*, *Xac* and *Xcc* [34, 35] harbors a XopL with 3 of 7 conserved PRR prolines (aligning with P133, 134, and 149 of XopL$_{Xe}$) and has relatively more basic amino acids in the PRR/α than XopL$_{Xcc}$ (15.8% vs. 11.9%). While at this point it is not known whether XopL from *X. hyacinthi* associates with MTs, it appears that an ancestral form of the MT binding region was present in the common ancestor of *Xcc* and *Xe*, a characteristic that was likely lost as *Xcc* diverged.

So far, differences in XopL MT-binding ability are not obviously correlated to specific modes of pathogen entry or to the plant host species, as both *Xac* and *Xe* prefer to enter through stomata and proliferate locally while *Xoo* and *Xcc* enter primarily through hydathodes (although hydathode architecture of plant species differs) and move systemically through the plant via the xylem [2]. Xanthomonads typically have limited host ranges, and interactions with their hosts shape effector repertoires and activities, applying strong selective pressure for the evolution of T3Es that subvert effector-triggered immune responses [2].

## XopL MT-binding mimics known eukaryotic MAPs

The presence of a PRR close to a region rich in basic amino acids, as in XopL proteins from xanthomonads, is reminiscent to the structure of known eukaryotic MT binding proteins (MAPs). PRRs in human MAPs, such as tau [27] and MAP2 [28] are required to stabilize protein-MT associations but are not sufficient for binding. In the case of tau, the PRR must act in concert with neighboring repeat regions harboring basic amino acids, with both protein regions interacting to facilitate MT binding [23, 36, 37]. It appears that XopL likely uses a similar mode for its association with MTs. This hypothesis is based on our finding that XopL derivatives lacking the PRR (αLRR_XL$_m$) are not MT-localized, nor are derivatives lacking the positively charged α region (NT), but together these regions are sufficient to bind MTs. It is worth noting, that while bacteria also possess tubulin homologs that assemble into filaments, bacterial tubulins lack acidic residues at the C-terminus that facilitate electrostatic interactions with kinesin motor proteins and MAPs [38]. Consistent with this, bacteria do not typically possess MAP homologs [39]. As far as we are aware, this is the first description of a MAP-like mechanism for MT binding described for any T3E. We suppose that XopL mimics features of both eukaryotic E3 ligases and MAPs.

## A model for XopL interaction with microtubule-associated substrates

Agroinfection, the transient protein expression system chosen for this study, provided the ideal platform for comparative analyses of XopL homologs from different *Xanthomonas* species, facilitating both the visualization of XopL localization in plant cells coupled with a macroscopic read-out for one of XopL's activities, plant cell death. While it must be considered that during an infection with *Xanthomonas* XopL protein concentrations are much lower, it appears unlikely that a decrease in protein level would alter the ability of XopL to associate with MTs, which occurs both *in vitro* (with isolated MTs) and *in planta*. Taken together, the results of this study support a model in which XopL associates with MTs through the NTα region. Thus, the LRR, the predicted protein interaction domain, remains free for interactions with MT-associated substrates.

## Interactions of XopL$_{Xe}$ at the microtubule induce plant cell death

In the case of agroinfection-mediated expression of XopL$_{Xe}$ in plant tissue, current data suggest that the modification of MT-localized targets, rather than MT disruption, triggers plant cell death. This is supported by the finding that no cell death occurs when MTs in plant cells are disrupted by other proteins, as shown here. *A. thaliana* KTN1 and PHS1, while destroying the MT network by 2 dpi, do not trigger macroscopic cell death, in contrast to XopL$_{Xe}$. Furthermore, XopL$_{Xe}$ derivatives that are unable to bind MTs also fail to cause cell death, and co-expression of XopL mutants with KTN1 or PHS1 did not restore cell death. This suggests that there is an alternative, MT-associated, E3 ligase-dependent mechanism of plant cell death induction. Whether the cell death evaluated here is related to XopL's virulence function or is the result of plant recognition and the initiation of programmed cell death triggered by XopL activity at MTs is enigmatic at this point.

## Are MTs essential for autophagic flux?

Autophagy is a conserved eukaryotic degradation pathway which manages the content of the cytoplasm, ridding the cell of proteins, nucleic acids, faulty organelles and lipids, either through selective mechanisms or in bulk [40]. Cellular content is enclosed within a double membrane to form a vesicle (autophagosome), which is transported to the vacuole (in plants) or to lysosomes (animals) where its contents are degraded and recycled [40]. Autophagy plays a complex role in plant immunity and is a common target of T3Es from pathogenic bacteria, which either enhance or suppress this process [41]. XopL$_{Xe}$ was recently reported to co-localize *in planta* to punctate structures with autophagy components AUTOPHAGY-RELATED GENE 8 (ATG8) and Joka2 and to inhibit autophagic flux [13]. MT localization of XopL is particularly interesting in this context as there are several lines of evidence to suggest that autophagy and MT dynamics are closely linked. In fact, Arabidopsis ATG8 isoforms are similar in sequence to known mammalian MT-associated proteins (MAPs) and are capable of direct MT binding *in vitro* and *in planta* [42, 43]. Tobacco ATG6, a component important for autophagosome assembly, also localizes to MTs and interacts with both β- and α-tubulin [44]. Most interesting is the finding that chemical or silencing approaches that decrease tubulin abundance or alter dynamics of MTs, inhibit stress-induced autophagy in both plant and animal cells [44, 45]. Taken together, this has led authors to speculate about the role of MTs in every aspect of autophagy from autophagosome biogenesis to transport or docking with the vacuole [42, 44, 45]. This brings about the question of whether the reported manipulation of autophagy by XopL$_{Xe}$ [13] depends on its MT localization. Furthermore, one must consider whether the strong inhibitory effect of XopL$_{Xe}$ on autophagic flux observed after agroinfection of *N. benthamiana* [13] is the result of MT destabilization.

## Implications of MT manipulation by bacterial pathogens

There are many examples of animal bacterial pathogens that target MTs and MAPs. Human pathogens *Shigella flexneri* and *Escherichia coli* employ T3Es VirA and EspG to depolymerize microtubules at invasion sites to ease bacterial entry into host cells [46–49]. In contrast, *Salmonella* species deliver multiple T3Es to modulate MT-mediated vesicle cycling [49]. In addition to T3Es, animal pathogenic bacteria secrete many factors and toxins that target microtubules to facilitate entry, movement and colonization of cells [49]. *Xanthomonas*, by contrast, does not enter the cells of host plants, but colonizes the spaces between cells, suggesting a different function for MT modulation. Furthermore, there are fundamental differences in the MT function and regulation between plant and animal cells [50, 51]. Therefore, specific efforts must be made to understand how phytopathogens exploit MT-related cellular processes to their benefit.

Only a handful of T3Es from phytopathogenic bacteria have been identified that manipulate plant MTs or MAPs. One example is the *Pseudomonas syringae* effector HopZ1a which directly binds MTs [52] and alters MT dynamics in *Arabidopsis thaliana*, acetylating both tubulin [52] and the MT-stabilizing kinesin HINKEL [53]. Other T3Es target MAPs, such as MAP65-1 which is targeted by the *P. syringae* T3E HopE1 [54] and ACIP, targeted by the *Xe* effector AvrBsT [55]. In case of HopE1 and HopZ1a the impact on MTs and MAPs was linked with a decrease in the secretion of antimicrobial proteins and callose deposition [52, 54], but so far the exact role of MTs in these processes is unclear. Evidence is emerging that suggests that MTs play a role in signal transduction during plant immune responses [56], and as already mentioned, MTs play an unspecified role(s) in autophagy. Although the targeting of MT-related processes by T3Es points to their importance in determining the outcome of plant-pathogen interactions, there is still much ambiguity surrounding the exact role of MTs and MAPs in plant defense. In this regard the future study of effectors, including $XopL_{Xe}$, and their targets may serve to gain further insight into MT-dependent defense responses and answer questions surrounding fundamental plant processes.

## Materials and methods

### Plant material and growth conditions

Wild-type *Nicotiana benthamiana* and stably expressing GFP labeled α-tubulin (GFP-At-TUA6; [21]) plants were grown in a greenhouse supplemented with light emitted by IP65 lamps (Philips) equipped with Agro 400 W bulbs (SON-T). Light intensity was approximately 130–150 $\mu E/m^2$ x s with light supplementation provided when light was below 100 $\mu E/m^2$ x s. Day length was 16 h at 26˚C with a relative humidity of 60%, and the temperature at night was 19˚C with a relative humidity of 40%.

### Agrobacterium strains and inoculations

*A. tumefaciens* strain GV3101 (pMP90) was used for all expression assays in this study. Bacteria were grown in liquid yeast extract broth (YEB; Vervliet et al., 1975) supplemented with 100 ug/mL rifampicin, 15 ug/mL gentamicin, and appropriate antibiotics for plasmid selection (pGGA plasmids = 100 μg/mL spectinomycin, pICH47751 = 100 μg/mL carbenicillin), and cultivated in a tube rotator at 30˚C overnight. Cells were harvested via centrifugation, re-suspended in *A. tumefaciens* infiltration medium (AIM = 10 mM MgCl2, 5 mM MES, pH 5.3 and 150 μM acetosyringone) and incubated at room temperature for 1 h. Bacteria were adjusted to an $OD_{600}$ = 0.4 of 0.8 prior to the inoculation of *N. benthamiana* lower leaves using a needle-less syringe.

## Taxol treatments

Treatments with Paclitaxel (taxol) (Cytoskeleton inc. Cat. # TXD01) were administered at 4 hpi with *A. tumefaciens* strains facilitating the expression of XopL$_{Xe}$, XopL$_m$ or the mCherry control protein fusions. Paxlitaxel was dissolved in DMSO to a concentration of 1 mM then diluted in AIM without acetosyringone to a concentration of 20 μmol and administered to *A. tumefaciens* inoculation spots via a needleless syringe. DMSO controls were diluted in the same way. Microscopy was performed at 2 dpi unless otherwise stated.

## Cloning

Unless otherwise stated XopL$_{Xe}$ constructs were amplified from plasmids generated by Singer et al., [12]. XopL$_{Xcc}$ was PCR-amplified directly from *Xcc* 8004 DNA. Derivatives of *A. thaliana* PHS1 (AT5G23720) and KTN1 (AT1G80350) were cloned from *A. thaliana* cDNA. KTN1 was amplified with primers harboring attB sites to facilitate Gateway cloning (Thermo Fisher Scientific). BP clonase II enzyme mix was used for a BP reaction into pDONR221. The pDONR221-KTN1 plasmid was then used as a template for further PCR reactions. PCR reactions on the templates described, used primers with *BsaI* or *BpiI* recognition sites to generate appropriate fragments for GoldenGate cloning [57].

For cloning of XopLs for UbiGate experiments and selected *in planta* assays XopLs from *Xoo* PX099a, *Xac* 306, *Xcc* 8004 and *Xe* 85–10 strains were synthesized as gene fragments with appropriate *BpiI* sites (GeneArt gene synthesis by Thermo Scientific; codon-optimized for *N. benthamiana*) and subcloned as level 0 CDS1ns modules (in pAGM1287) for use in the MoClo cloning system [58]. CDS1ns modules were used for the assembly of UbiGate plasmids or plant expression vectors. For *in planta* expression of XopL, CDS1ns modules were assembled in combination with the '35S long' promoter (pICH51266), a C-terminal mCherry tag (pICSL50004), and a 35S terminator (pICH41414) module into the plant expression vector pICH47751 (Amp$^R$) [58]. The mCherry control was assembled using the same 35S promoter and terminator modules but with the mCherry module pICSL80007 [58]. For UbiGate, codon-optimized XopL CDS1ns modules were assembled into pET-28GG (Kan$^R$) under control of the T7 promoter [20]. Final pET-28GG plasmids were assembled from modules published by [20]: pGG-193 (His-tagged *A. thaliana UBIQUITIN 10*) and pGG-194 (untagged *A. thaliana UBIQUITIN-ACTIVATING ENZYME 1*), in combination with modules generated in this study; UG36 (GST-tagged UBIQUITIN CONJUGATING ENYZME 28; assembled from pUG-UBC28 [20] and pAGM18012 [20]), and XopL modules (XopLs with C-terminal 4x Myc tags; assembled from XopL CDS1ns and pICSL50010 [58]).

For cloning of non-codon-optimized XopLs, derivatives and *A. thaliana* genes for expression *in planta*, a set of GoldenGate-compatible vectors of the pGG system (Schulze et al., 2012) were used. As described in Schulze et al. [59] all pGG vectors are preassembled, using the pBGWFS7 backbone with spectinomycin resistance for selection in *Agrobacterium*, with 35S promoter and terminator plus N- or C-terminal fluorescence tags. All proline exchange mutations were introduced into the XopL wild type or E3 ligase mutant sequences via PCR-based single/multiple-site mutagenesis [60]. Unless otherwise stated, *BsaI*/T4-ligase cut-ligations facilitated cloning into all vectors, including an extra ligation step to repair any internal *BsaI* sites [57]. For overexpression of SUMO-tagged recombinant proteins in *E. coli* restriction-free cloning [61] was used to introduce strep-XopL, and strep alone into the pETSUMO plasmid (Thermo Fisher). A 352 bp fragment (5453–5804 bp) directly following the SUMO tag up to the *BamHI* site was replaced with strep-XopL or strep alone (control construct) to create 6xHIS-SUMO-strep-XopL and 6xHIS-SUMO-strep constructs. RF-Cloning.org [62] was used to generate primers. Primer sequences are available upon request.

## Confocal microscopy

Confocal microscopy was done using a Zeiss LSM 780, inverted AxioObserver, with a 63x water lens (C-apachromatx63/1.20 W Korr M27) or a Leica STELLARIS 8 inverted fluorescent microscope, with a 40x water lens (HC PL APO40x/1.10). In case of the Zeiss LSM780 laser lines of 488 nm and 561 nm were used to excite GFP and mCherry fluorophores, respectively, and channels were defined from 493–523 nm (GFP), 588–632 nm (mCherry) and 684–721 nm (chlorophyll). ZenBlack software controlled the microscope and image capture. In case of the Leica, fluorophores were excited with a white light laser, and channels were defined the same as described for Zeiss microscopy. LasX software controlled the microscope and image capture.

## Image processing and analysis with Fiji

Confocal images were collected as z-stacks (.czi or.lif files), and images were processed using the 'Z-project' function of Fiji [25, 26] to create an extended depth of focus. Fiji [25, 26] was also used to quantify MT association of fusion proteins and MT destabilization. For these experiments, mCherry labeled fusion proteins were expressed in GFP-TUA6 plants. To quantify MT labeling, MTs were located and marked with the line tool in the GFP channel; mean gray values were measured in the mCherry channel, thus quantifying the brightness of the mCherry signal in the location of the filament. To quantify the cytosolic background, a neighboring region parallel to the MT was also measured. %MT fluorescence was calculated by dividing MT brightness by the cytosolic background. %MT measurements were collected at random along 10 MTs in 15 cells per treatment. To quantify the number of MTs remaining in cells expressing fusion proteins, lines of 30 μm were drawn at random in cell lobes (avoiding the nucleus and cell boundaries) and MTs crossing this line were quantified using the 'Cell Counter' tool. One line was drawn per cell and at least 10 cells were measured per treatment. SigmaPlot was used for the statistical analysis of all data.

## *E. coli* growth and overexpression

Chemically competent *E.coli* BL21 DE3 RIL cells were freshly transformed with plasmid DNA and cells were cultivated in 50 mL liquid lysogeny broth (LB; [63]) medium containing 50 μg/mL kanamycin overnight at 37˚C (shaking at 140 rpm). The following day a main culture of 20 mL was started for UbiGate assays or 500 mL for SUMO tagged protein overexpression and purification. Cultures were inoculated at a starting $OD_{600nm}$ of 0.05. At an $OD_{600}$ of 0.6–0.8, at which point protein expression was induced with isopropyl β-D-1-thiogalactopyranoside (1 mM final concentration) and 99% EtOH (3%). For UbiGate, cultures were incubated in a shaker (140 rpm) at 28˚C for 3 hours. For SUMO-tagged XopL and control overexpression cells were incubated in a shaker (140 rpm) overnight at 16˚C. Bacterial cells were harvested using a tabletop Eppendorf centrifuge (5810R) at 4000 xg with a swing-out rotor (A-4-62), and cells were frozen at -80˚C until samples were processed.

## UbiGate assays

The UbiGate protocol was modified from Kowarschik et al. [20]. *E. coli* cells were resuspended in 400 μL 1x PBS buffer and adjusted to an $OD_{600nm}$ of 13 in a volume of 100 μL to normalize for differences. Cell suspensions were treated with 1 μL lysozyme (10 mg/mL stock) for 15 min and 4x freeze-thaw (samples were transferred between liquid nitrogen and a 30˚C heating block). Lysates were then treated with 2 μL RNase and DNase (stock 1 mg/mL) and incubated

for 30 minutes. 100 μL of 2x Laemmli buffer was added and samples were boiled at 95˚C for 10 minutes. Samples were stored at -20˚C until analysis.

## Purification of proteins and MT spin-down assays

*E. coli* cells expressing 6xHIS-strep-SUMO-XopL and the tag only control (6xHIS-strep-SUMO) were resuspended in 25 mL of PBS + PMSF (1 mM), and subjected to a French press (1000 psi, 3 cycles). All following procedures were conducted at 4˚C. Cell lysates were cleared via centrifugation (30 min, 4000 rpm in a Beckman Coulter Avanti J-25 centrifuge with a F15-6x100y rotor). Supernatants were subjected to ultracentrifugation (60 min, 28, 000 rpm in a Beckman Coulter Optima MAX Ultracentrifuge with a SW-32 Ti rotor) to sediment membranes, and supernatants were transferred to a 100 mL bottle and topped up to 100 mL with Ni-column binding buffer (20 mM HEPES (pH 7.5), 500 mM NaCl, 20 mM Imidazole) and transferred to a gravity flow Ni-column via peristaltic pump (50 μL/min). Econo-Pac Chromatography Columns (20 mL; BioRad) packed with $Ni^{2+}$ nitrilotriacetic acid (NTA)-sepharose (GE Healthcare) were used. The column was washed with 5 matrix volumes of binding buffer, followed by an elution step (Elution buffer: 20 mM HEPES (pH 7.5), 500 mM NaCl, 500 mM imidazole); the purified protein was detected in the flow-through via Western Blot.

To prepare MT binding assays the buffer was exchanged to eliminate salt and imidazole, which could interfere with assays. For this, purified protein was loaded into cellulose membrane tubing for overnight dialysis (3.5 kDa cutoff) into four different buffers (volume was 1000x that in the membrane) for a solubility test (details in S7A Fig). Due to low XopL solubility in buffers without salt (required for the spin-down assay), the SUMO tag was not cleaved off. XopL was incubated in HEPES buffer containing 80 mM HEPES, 5 mM MgCl2 and 250 mM of aminocaproic acid. Immediately following dialysis, protein samples were subjected to ultracentrifugation to remove aggregates (55, 000 rpm, 20 min, 4˚C). MT assays were performed using the MT Binding Protein Spin-down kit from Cytoskeleton, Inc. (Cat. # BK029), according to manufacturer instructions. Samples were stored in Laemmli buffer [64] at -20˚C prior to analysis via SDS-PAGE and immunoblot or Coomassie staining. Protein amount was quantified using Fiji [25, 26].

## Protein preparation and western blot

For the analysis of *in planta* protein level, agroinfected plant material from two independent plants was ground in liquid nitrogen, resuspended in 120 μL of 2x Laemmli, boiled for 10 min at 95˚C, and stored at -20˚C.

Proteins were analyzed via SDS-PAGE followed by staining with Coomassie Brilliant Blue [65], or immunoblot using the following monoclonal antibodies: α-Strep (1:5000, Sigma-Aldrich), α-P4D1 at (1:5000, Millipore), α-GFP (1:2000, Roche diagnostics) at 1:2000, α-c-Myc (1:2000, Roche diagnostics), DM1A (1:1000, Sigma-Aldrich), or the following polyclonal antibodies: α-mCherry (1:2000, Abcam) and α-GST (1:10, 000, Sigma-Aldrich). Secondary α-rabbit, α-mouse and α-goat antibodies (GE Healthcare) were coupled with horseradish peroxidase for visualization of proteins by enhanced chemiluminescence.

## Protein sequence and structural alignments

Using the phylogenetic trees of the *Xanthomonas* species [34, 35] as guides, NCBI blastp with the αLRR_XL amino acid sequence of XopL$_{Xe}$ were used to identify XopLs from representative strains throughout the genus. Multiple sequence alignments were conducted using Geneious Prime, Clustal Omega 1.2.2.

XopL$_{Xe}$ and XopL$_{Xcc}$ 3D structures were generated from full length XopL sequences utilizing AlphaFold2 [66, 67]. Protein structural alignments were conducted using Geneious Prime.

Electrostatic surface calculations were performed using PyMOL's APBS Electrostatics plugin [68] and results were visualized using PyMOL software (www.pymol.org).

## Supporting information

**S1 Fig. Ubigate of XopL$_{Xe}$ variants previously analysed by Singer et al.** [12]. (A) Schematic of the UbiGate plasmid used to test for E3 ligase activity of XopL proteins. Components of the *A. thaliana* ubiquitination machinery (affinity-tagged for detection by western blot), *UBIQUI-TIN 10* (*HIS*:*Ub*), *UBIQUITIN ACTIVATING ENZYME 1* (*UBA1*) and *UBIQUITIN CONJU-GATING ENZYME 28* (*GST*:*UBC28*) were expressed from a single, IPTG-inducible plasmid together with XopL coding sequences. Each translational unit was equipped with an independent ribosome binding site (red). (B-D) Western blot analysis of protein extracts isolated from *E. coli* expressing different XopL variants. Controls were samples lacking the E3 ligase (E3-), the E2 enzyme (E2-) or ubiquitin (Ub). The XopL protein variant tested is indicated above the designated lanes. (B) Ubiquitin was detected using the P4D1 antibody. Polyubiquitin chains are indicated by a black line. E2-ubiquitin (Ub+E2) conjugates are visible in the E3- control sample, XopL$_m$ and NTαLRR samples (size indicated by a purple line). (C) C-terminally tagged XopL proteins were detected with a myc-specific antibody (expected size indicated with '*'). In some cases, autoubiquitination is detectable (Ub+XopL). (D) N-terminally-tagged E2 enzyme was detected using a GST-specific antibody (expected size indicated with '*'). E2-ubiquitin conjugates are indicated with a black line. Samples were run on different gels to clearly visualize ubiquitin and other proteins. Protein mass is expressed in kDa.
(TIFF)

**S2 Fig. mCherry does not co-localize with MTs.** Confocal microscopy of a lower epidermal cell of a GFP-TUA6 (labels MTs) transgenic *N. benthamiana* leaf. The leaf was agroinfected to express mCherry (OD$_{600}$ of 0.4). Samples were harvested for microscopy 2 dpi. (A) GFP-labeled microtubules are in white, (B) mCherry in magenta, (C) is a merged image of (A) and (B). Plastids are colored in cyan. Scale bars are 20 μm. (D-F) are magnified from images (A), (B) and (C), respectively. Scale bars are 5 μm. The yellow line in (F) shows the location of the intensity plot measurement depicted in (G). (G) Fluorescence intensity plot across two MTs (visible as two distinct peaks in black). mCherry and chlorophyll intensity profiles were included as negative controls.
(TIFF)

**S3 Fig. Taxol partially rescues MTs from XopL$_{Xe}$.** Confocal microscopy of lower epidermal cells of GFP-TUA6 (labels MTs) stable transgenic *N. benthamiana* leaves. Leaves were agroin-fected (OD$_{600}$ of 0.4) to express (A-C) XopL$_{Xe}$-mCherry, (D-F) XopL$_m$-mCherry and (K-M) mCherry and treated with the MT stabilizing chemical taxol (solved in DMSO) at 4 hpi. Samples were harvested for microscopy 2 dpi. The GFP channel is visible in white (labeled MTs) and the mCherry channel in magenta. Plastids are in cyan; 'n' labels nuclei. Scale bars are 20 μm. (G), (H), (I), (J) Are magnified images from (A), (B), (D) and (E), respectively (area magnified is outline with a white box). Scale bars in (G-J) are 5 μm. Examples of MTs are labeled with white arrows.
(TIFF)

**S4 Fig. DMSO control infiltrations into XopL$_{Xe}$-expressing tissue did not rescue MTs.** Confocal microscopy of lower epidermal cells of GFP-TUA6 (labels MTs) stable transgenic *N. benthamiana* leaves. Leaves were agroinfected (OD$_{600}$ of 0.4) to express (A-C) XopL$_{Xe}$-mCherry, (D-F) XopL$_m$-mCherry and (K-M) mCherry which was then treated with DMSO at 4 hpi. Samples were harvested for microscopy 2 dpi. The GFP channel is visible in white

(labeled MTs), the mCherry channel in magenta. Plastids are visible in cyan. Scale bars are 20 μm.
(TIFF)

**S5 Fig. XopL$_{Xoo}$ and XopL$_{Xac}$ localize to MTs.** Confocal microscopy of lower epidermal cells of GFP-TUA6 stable transgenic *N. benthamiana* leaves. Leaves were agroinfected (OD$_{600}$ of 0.4) to express synthesized (codon-optimized) (A) XopL$_{Xac}$ and (B) XopL$_{Xoo}$ translationally fused to a C-terminal mCherry. (A) and (B) are zoomed-out versions of cells depicted in Fig 4G and Fig 4J, respectively. mCherry-tagged XopLs are visible in magenta and the GFP channel is not shown here, 'n' marks the nucleus, white arrows show example MTs. Scale bars are 20 μm. Insets are magnifications of the nuclei (scale bar is 10 μm). (C) Plant reactions to codon-optimized XopLs were monitored at 6 dpi.
(TIFF)

**S6 Fig. Clustal Omega multiple sequence alignment of XopLs from *Xoo* PX099A (accession ACD59771), *Xcc* 8004 (WP_011038250), *Xac* 306 (AAM37935) and *Xe* 85–10 (CAJ24951).** Amino acids are colored based on polarity (Geneious Prime). Acidic amino acids in red, basic in blue, and fuchsia highlights prolines. Domains are marked as follows: the proline-rich region (PRR) in gray, the three alpha helices in yellow, and the 9 LRR repeats and XL box in pink.
(TIFF)

**S7 Fig. Optimization of XopL$_{Xe}$ solubility for MT co-sedimentation assay and MT-binding assay controls.** (A) Coomassie-stained 10% polyacrylamide gel showing the optimization of XopL$_{Xe}$ solubility under low salt conditions. HEPES buffer without salt at pH 6.9 or 5 was supplemented with 250 mM aminocaproic acid (+) to maximize solubility of 6xHis-StrepII-SUMO-XopL (88.1 kDa) for MT co-sedimentation assays. (B) Coomassie-stained 10% polyacrylamide gel of co-sedimentation assay. Supernatant fractions are marked 'S' and pellet fractions 'P'. Samples with MTs are marked '+' and without by '-'. Bands at the expected sizes for the BSA negative control (68 kDa; lanes 3–6), the bovine MAP fraction (a mixture of MT-associated proteins eluted from bovine brain MTs) positive control (250 kDa; lanes 7–10), and XopL (88.1 kDa; lanes 11–14) are marked with '*'. A MT only sample served as a control (lanes 1–2). Double tubulin bands marked with '<'. The assay was repeated 2 times with the kit controls, assays with varying XopL concentrations were repeated more than 3 times with comparable results.
(TIFF)

**S8 Fig. MT association is correlated with MTs disassembly.** Linear regression comparing MT binding ability of XopL$_m$ derivatives with MT number remaining after expression of XopL$_{Xe}$ derivatives. Each data point represents the MT association of a given E3 ligase mutant variant (graphed in Fig 5C) plotted against the MT number remaining after expression of the corresponding E3 ligase-active version (graphed in Fig 5D). Each data point is labeled with the derivative name. The line of best fit is blue, and the equation of the line is displayed in the upper right (Linear Regression, $R^2 = 0.762$, F [1, 7] = 22.35, P = 0.002).
(TIFF)

**S9 Fig. Western blot analysis of XopL$_{Xe}$ NTα truncations.** Protein extracts isolated 2 dpi from the experiment depicted in Fig 7. Signals were detected with GFP-specific antibody (*). The left side of the blot shows protein mass in kDa.
(TIFF)

**S10 Fig. KTN1 and PHS1 destroy MTs.** Confocal microscopy of lower epidermal cells of GFP-TUA6 stable transgenic *N. benthamiana* leaves. Leaves were agroinfected ($OD_{600}$ of 0.8) to express (A) mCherry, (B) KTN1, (C) PHS1 185-700aa ($PHS_{185-700}$) or (D) PHS1 1-700aa ($PHS_{1-700}$) tagged with GFP. Samples were harvested for microscopy at 2 dpi. Images show the GFP channel, where MTs (GFP-TUA6 labeled) are typically visible (i.e., panel A). Scale bars are 20 μm. (E-H) are zoomed in versions of (A-D) respectively. (I) Cell death quantification via red fluorescence scanning of agroinfected *N. benthamiana* leaves. Tissue co-expressing $XopL_{Xe}$ (purple) or non-MT-binding derivatives ($ex2_{127/130}$; blue and αLRR_XL; orange) together with GFP or MT-disrupting proteins KTN1 and $PHS1_{185-700}$ was monitored for cell death 5 dpi. Boxes represent first to third quartiles, the median is marked by a horizontal line and whiskers show the distribution of remaining data points. Treatments that were significantly different than $XopL_{Xe}$+ GFP co-inoculations are marked with asterisks (* = $p<0.05$, ** = $p<0.01$, *** = $p>0.001$; One Way Analysis of Variance on Ranks, Bonferroni post-hoc test). (TIFF)

**S11 Fig. Clustal Omega multiple sequence alignment of select XopL homologs from the *Xanthomonas* genus.** XopL protein sequences from 24 strains were aligned. The strain of origin is listed on the left-hand side with the NCBI accession number of the XopL protein sequence. Amino acids are colored based on polarity (Geneious Prime). Acidic amino acids in red, basic in blue, and fuchsia highlights prolines. The $XopL_{Xe}$ proline-rich region (PRR) is in gray, the alpha α region (α-helices 1, 2 and 3) in yellow, and the beginning of the LRR domain (visible as 'LRR1') in light pink. The sequence logo above the alignment shows sequence conservation at specific positions. (TIFF)

## Acknowledgments

We are grateful to Bianca Rosinsky, Marina Schultze, and Carola Kretschmer for excellent technical assistance, Ralf-Bernd Klösgen for help with MT spin-down assays, Karl Oparka for providing GFP-TUA6 *N. benthamiana* seeds, Gina Stamm for the KTN1 subcloning, Sabine Thieme, Oliver Nagel, Wiebke Kummer, Norman Adlung and Lennart Wirthmüller for experimental advice, Marco Trujillo for generously supplying us with the UbiGate cloning system, Katharina Bürstenbinder, Sebastian Schornack, Johannes Stuttmann and Martin Schattat for advice and discussion. A special thanks to Tina Romeis for reviewing the manuscript and for supporting key experiments.

## Author Contributions

**Conceptualization:** Vinzenz Handrick, Sarah Zinecker, Matthias Reimers, Ulla Bonas, Jessica Lee Erickson.

**Data curation:** Simon Ortmann, Jolina Marx.

**Formal analysis:** Simon Ortmann, Jolina Marx, Vinzenz Handrick, Jessica Lee Erickson.

**Funding acquisition:** Ulla Bonas, Jessica Lee Erickson.

**Investigation:** Simon Ortmann, Jolina Marx, Christina Lampe, Sarah Zinecker, Jessica Lee Erickson.

**Project administration:** Christina Lampe, Tim-Martin Ehnert.

**Supervision:** Jessica Lee Erickson.

**Writing – original draft:** Jessica Lee Erickson.

**Writing – review & editing:** Ulla Bonas, Jessica Lee Erickson.

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
