## [Decision Letter · Decision Letter 0]

23 Apr 2023

Dear Dr. Erickson,

Thank you very much for submitting your manuscript "A conserved microtubule-binding region in Xanthomonas XopL is indispensable for induced plant cell death reactions." for consideration at PLOS Pathogens. As with all papers reviewed by the journal, your manuscript was reviewed by members of the editorial board and by several independent reviewers. Overall, the reviewers were quite positive about the quality and significance of the work. We invite the resubmission of a significantly-revised version that takes into account the valuable suggestions of reviewers #1 and #2 (comments from all three reviewers are below this email).

We cannot make any decision about publication until we have seen the revised manuscript and your response to the reviewers' comments. Your revised manuscript is also likely to be sent to reviewers for further evaluation.

Sincerely,

David Mackey

Academic Editor

PLOS Pathogens

Shou-Wei Ding

Section Editor

PLOS Pathogens

Kasturi Haldar

Editor-in-Chief

PLOS Pathogens

orcid.org/0000-0001-5065-158X

Michael Malim

Editor-in-Chief

PLOS Pathogens

orcid.org/0000-0002-7699-2064

Reviewer's Responses to Questions

**Part I - Summary**

Reviewer #1: In this report the authors used a comparison between polymorphic alleles of the conserved Xanthomonas effector XopL to investigate the role of its subcellular localization in host cell in regard with its functionality.

Using the well-established transient expression system in the model N. benthamiana and live imaging by confocal microscopy, the authors show that several natural variants of XopL can associate with microtubules (MTs) in plant cell and destabilize the MT network. Although XopL appears to directly bind MTs, it does not degrade tubulin, leading the authors to hypothesis that XopL E3-ligase activity must target MT-associated proteins to disturb the MT network. By sequence comparison with the XopLXcc variant that does not associate with MT and using XopL truncations, the authors could identify an N-terminal region of XopL encompassing a proline-rich region (PRR) and a short alpha-helix sufficient for XopL MT-binding. This region bears structural homologies with animals microtubule-associated protein MT-binding domains and is sufficient to confer MT-binding activity to the XopLXcc variant. The importance of the PRR for MT-binding and MT network destabilization was further confirmed by mutagenesis of the proline residues. Lastly, the authors explore the link between XopL MT-binding activity and the induction of cell death, used here as a proxy for XopL function in plants. XopL MT-binding is not sufficient to trigger plant cell death, as the E3-ligase activity is also necessary for this response. The MT network destabilization is not sufficient to induce cell death, as shown by expressing MT network destabilizers, unrelated to XopL. However, XopL-induced cell death correlates with its ability to bind MT when it carries the functional E3 ligase domain, as shown for example with the fusion of the XopLXe-NTa (mediating MT-binding) with the XopLXcc-LRR and XL box domains.

Overall, the study was well designed and conducted with appropriate methods and controls. I commend the authors for the careful interpretation of their data in light of previous studies and the discussion that clearly put in perspectives their findings.

Reviewer #2: This interesting and relevant study does a deep dive into the molecular mechanisms that underpin T3SS effector function in plants. Previously, this group demonstrated that the E3 ligase, XopL, from Xanthamonas spp. associated with the microtubule (MT) cytoskeleton in the absence of E3 ligase activity and may is involved in chloroplast stromule dynamics. Here, by exploring XopL activities, sequence conservation, and domain architecture across Xanthamonas species, they find that XopLxcc from Xanthamonas campestris campestris is unable to bind associate with microtubules. Through deletion analysis, mutagenesis, and domain swap experiments, they demonstrate that MT binding is conferred by a proline-rich tract and the NT��region it is embedded within. Further, the authors show that in planta, microtubules are disrupted by XopLxe and that this perturbation requires E3 ligase activity. Cell death also correlates with the combination of MT-binding and destruction and E3 ligase activity; however, disruption of MTs by overexpression of MAPs is not sufficient to initiate cell death. In a simple pulldown experiment, the authors demonstrate the ability to bind MT in vitro; however, the molecular mechanism of MT destabilization in planta is likely indirect and requires further exploration. I find this body of work to be impressive, experiments carefully conducted and adequately controlled, and the results critically interpreted.

Reviewer #3: The manuscript （PPATHOGENS-D-23-00407）， “A conserved microtubule-binding region in Xanthomonas XopL is indispensable for induced plant cell death reactions.”， written by Ortmann et al., described their novel finding that the ancestral effector XopL displays bacterial species dependent differences in their sub-cellular localization and plant cell death reactions. It is very interesting that XopL from X. euvesicatoria (XopLXe) directly associates with plant microtubules (MTs) and causes strong cell death in agroinfection assays in N. benthamiana. This event is not associated with XopLXcc which come from X. campestris pv. campestris that fails to localize to MTs and to cause plant cell death. Thus，the authors further confirmed that a proline-rich-region (PRR)/α-helical region is important for MT localization. The MT-localized XopL activity is required for plant cell death reactions. Best to my knowledge, this is the first report that a T3E within the context of a genus rather than a single species can shed light on how effector localization is linked to biochemical activity.

The data are sufficient to support the statement, the manuscript is well organized and written. I would like it accept for publication in this lovely journal.

**Part II – Major Issues: Key Experiments Required for Acceptance**

Reviewer #1: A main concern (and to generalize the authors findings) is the cell death-inducing activity of XopLXoo and XopLXac. The comparison of XopLXe with these two MT-binding variants could be used to further support the NTalpha region features required for MT-binding (in chimera with Xcc LRR and XL domains) and to clarify the link with cell-death inducing activity of MT-binding XopL variants.

Reviewer #2: A couple of suggestions for further improvement of these studies:

1) The in vitro microtubule binding studies shown in Fig. 6A should be conducted with a dose series and the data for XopL in the pellet curve fit to estimate a Kd value. Ideally, this would be done with a known MT side-binding protein as a positive control. Finally, binding of the truncated and/or mutated forms of XopL should be examined to demonstrate that the PRR in NT� is necessary and sufficient for MT binding.

2) They have established nice methods to quantify XopL-MT colocalization as well as the effects on microtubule numbers. The former should be applied to the experiments shown in Fig. 7 with the XopL truncations and domain swap constructs to further strengthen the conclusions derived from the observations. I would also suggest to move the excellent correlation analysis shown in Fig. S8 moved into the main text.

Reviewer #3: The data are validate to support the conclusion.

**Part III – Minor Issues: Editorial and Data Presentation Modifications**

Reviewer #1: As a general note about the presentation of the results, I recommend the authors to shorten and merge certain parts of the result sections. For example: the 4 XopL alleles E3 ligase activity; the 4 XopL alleles localization; the 4 XopL alleles induced cell-death… Although confocal images are space consuming, splitting controls and samples in different mains and supplemental figures equally splits the attention of readers.

P6l2: hints -> hint (variability and difference in cell death)

P6l12-22: move to introduction or discussion

P9l5: rephrase, XopL-fusion expression induced the chloroplast clustering (the XopL-fusion do not “displayed plastid clustering”)

P10l17, p25l23: change “solved” to dissolved in DMSO

P13l17: Singer et al referencing number?

P16l9: protein size and accumulation/stability were confirmed

P16l13: singular (-> XopL-induced decrease in MT abundance is correlated with MT affinity)

P17l19-20: rephrase “surface-exposed pocket” (?)

P20l15: XopLXac and XopLXoo did not trigger cell death in benthie (Figure S5) -> see main comment and add reference for Xac and Xoo cell death in benthies/other plants?

Figure 1 legend: indicate the expected size of the XopL variants (multiple bands for XopLXe and XopLXoo)

Figure 1 UbiGate control: was the XopLm construct tested with this method?

Figure 2: how is “high” or “low” level of XopLXe-GFP determined? Could the authors indicate the image acquisition conditions (i.e.: gain) used to clarify?

Figure 3 legend l3: change “Xopm-mCherry” to XopLm-mCherry

Figure 3 M: asterisks are not described in the legend and don’t appear to be necessary.

Figure 3: why not showing TUA6-GFP immunoblot?

Figure 4 A and B + legend: the boxes are outlined in black not in white (l6) and difficult to make out.

Figure 6 legend l7-8: replace 3x and 2x by three/ two times

Figure 6 D: the box plot for XopLXe is not on the graph…

Figure 6 E: no loading control?

Figure 8 C: this panel should be split into experiment shown in A and experiment shown in B as these were done independently.

Figure S6 and S11: change Xcv to Xe for the PRR box

Figure S10 I: add y-axis title (Cell death or mean gray value)

Reviewer #2: N/A

Reviewer #3: There are no minor issues for revision.

PLOS authors have the option to publish the peer review history of their article (what does this mean?). If published, this will include your full peer review and any attached files.

Reviewer #1: No

Reviewer #2: No

Reviewer #3: **Yes: **Gongyou Chen
---

## [Decision Letter · Decision Letter 1]

17 Jul 2023

Dear Dr. Erickson,

We are pleased to inform you that your manuscript 'A conserved microtubule-binding region in Xanthomonas XopL is indispensable for induced plant cell death reactions.' has been provisionally accepted for publication in PLOS Pathogens.

Best regards,

David Mackey

Academic Editor

PLOS Pathogens

Shou-Wei Ding

Section Editor

PLOS Pathogens

Kasturi Haldar

Editor-in-Chief

PLOS Pathogens

orcid.org/0000-0001-5065-158X

Michael Malim

Editor-in-Chief

PLOS Pathogens

orcid.org/0000-0002-7699-2064

Reviewer Comments (if any, and for reference):

Reviewer's Responses to Questions

**Part I - Summary**

Reviewer #1: The authors have addressed all the revierwers' questions/comments with care. This is a nice piece of work!

Reviewer #2: The authors have thoroughly and adequately addressed my concerns with this revision. I can recommend it for publication without further revision.

**Part II – Major Issues: Key Experiments Required for Acceptance**

Reviewer #1: no further comments

Reviewer #2: N/A

**Part III – Minor Issues: Editorial and Data Presentation Modifications**

Reviewer #1: (No Response)

Reviewer #2: N/A

PLOS authors have the option to publish the peer review history of their article (what does this mean?). If published, this will include your full peer review and any attached files.

Reviewer #1: No

Reviewer #2: No

---

## [Editor Report · Acceptance letter]

9 Aug 2023

Dear Dr. Erickson,

We are delighted to inform you that your manuscript, "A conserved microtubule-binding region in Xanthomonas XopL is indispensable for induced plant cell death reactions.," has been formally accepted for publication in PLOS Pathogens.

Best regards,

Kasturi Haldar

Editor-in-Chief

PLOS Pathogens

orcid.org/0000-0001-5065-158X

Michael Malim

Editor-in-Chief

PLOS Pathogens

orcid.org/0000-0002-7699-2064